# From Moments to Models: Graphon-Mixture Learning for Mixup and Contrastive Learning

**Ali Azizpour** [* 1]   **Reza Ramezanpour** [* 1]   **Santiago Segarra** [1]

## Abstract

Real-world graph datasets often arise from mixtures of populations, where graphs are generated by multiple distinct underlying distributions. In this work, we propose a unified framework that explicitly models graph data as a mixture of probabilistic graph generative models represented by graphons. To characterize and estimate these graphons, we leverage graph moments (motif densities) to cluster graphs generated from the same underlying model. We establish a novel theoretical guarantee, deriving a tighter bound showing that graphs sampled from structurally similar graphons exhibit similar motif densities with high probability. This result enables principled estimation of graphon mixture components. We show how incorporating estimated graphon mixture components enhances two widely used downstream paradigms: graph data augmentation via mixup and graph contrastive learning. By conditioning these methods on the underlying generative models, we develop graphon-mixture-aware mixup (GMAM) and model-aware graph contrastive learning (MGCL). Extensive experiments on both simulated and real-world datasets demonstrate strong empirical performance. In supervised learning, GMAM outperforms existing augmentation strategies, achieving new state-of-the-art accuracy on 6 out of 7 datasets. In unsupervised learning, MGCL performs competitively across seven benchmark datasets and achieves the lowest average rank overall.

---

[*]Equal contribution [1]Department of Electrical and Computer Engineering, Rice University, Houston, TX, USA. Correspondence to: Ali Azizpour <aa210@rice.edu>, Reza Ramezanpour <rr68@rice.edu>.

*Proceedings of the 43rd International Conference on Machine Learning*, Seoul, South Korea. PMLR 306, 2026. Copyright 2026 by the author(s).

## 1. Introduction

Graphs provide a powerful framework for modeling complex relational data across diverse domains, including social networks (Yang et al., 2021; Kumar et al., 2022), bioinformatics (Balaji et al., 2022; Zhang et al., 2021; Azizpour et al., 2024), and wireless systems (Zhao et al., 2023; Olshevskyi et al., 2025). A central goal in graph machine learning is to learn representations that capture the underlying principles governing these networks for tasks like graph classification (Rey et al., 2025). Graphons, or graph limits, have emerged as a foundational tool for this, offering a continuous, non-parametric generative model that describes the large-scale structure of graphs (Lovász, 2012; Borgs et al., 2008). By representing the latent process from which graphs are sampled, graphons enable principled analysis and data augmentation (Navarro & Segarra, 2022; Han et al., 2022).

Given their utility, the accurate and efficient estimation of graphons (Chan & Airoldi, 2014; Xu et al., 2021; Xia et al., 2023) from observed network data is a central problem. This has spurred the development of several modern estimation techniques. For instance, methods like Scalable Implicit Graphon Learning (SIGL) (Azizpour et al., 2025) leverage implicit neural representations and graph neural networks to learn a continuous graphon at arbitrary resolutions from large-scale graphs. Other recent work has shown that leveraging graph moments (subgraph counts) provides a computationally efficient path to scalable graphon estimation (Ramezanpour et al., 2025). However, a critical limitation of these approaches is that they are designed to estimate a single, unified graphon for the observed set of graphs. This assumption fails in real-world settings, where datasets frequently consist of a *mixture of populations*, with graphs generated from several distinct underlying distributions, and as a result, modeling a single graphon may not capture highly heterogeneous graph data (Ramezanpour et al., 2025).

This limitation also extends to modern representation learning paradigms, which often overlook the shared or distinct underlying models of different graphs. For example, graph augmentation techniques such as G-mixup (Han et al., 2022) assume a single graphon per class of data to serve as the latent representation for mixing. However, graphs within

the same class may arise from multiple generative processes, leading to heterogeneous latent spaces and semantically inconsistent augmentations. Similarly, in graph contrastive learning (You et al., 2020; 2021), each graph serves as an anchor, the positive sample whose representation is being learned, while all other graphs in the batch are treated as negatives. This neglects the possibility that some graphs originate from the same underlying generative model, making them false negatives and hindering representation quality.

To explicitly address this challenge, we introduce a unified framework that models graph data as a mixture of underlying graphons. We rely on the key insight that *densities of motifs* (empirical graph moments) serve as a powerful signature for the underlying generative model (Borgs et al., 2010). Building on foundational results, we establish a novel theoretical guarantee, showing that graphs sampled from graphons with a small cut distance will have similar motif densities with a higher probability. By exploiting these motif densities, we first partition the graph dataset into coherent clusters, where each cluster corresponds to a distinct component of the graphon mixture. This motif-aware partitioning allows us to disentangle the generative mechanisms at play and incorporate the graphon mixture in downstream graph-level tasks, whether supervised or unsupervised. Our main contributions are as follows.

- We introduce the first motif-based clustering approach that partitions graph datasets into coherent clusters, each corresponding to a distinct underlying graphon, thereby enabling model-aware graph-level learning.

- For supervised settings, we propose graphon-mixture-aware mixup (GMAM) for graph data augmentation, which performs mixing conditioned on the inferred graphon components within each class and enhances graph classification performance.

- For unsupervised learning, we develop a model-aware graph contrastive learning framework (MGCL) that leverages graphon-specific augmentations and model-aware negative sampling to improve representation quality.

**Conflict of Interest Disclosure**    The authors declare no competing interests.

## 2. Preliminary

### 2.1. Graphon

A graphon is defined as a bounded, symmetric, and measurable function $W : [0,1]^2 \rightarrow [0,1]$ (Lovász, 2012; Avella-Medina et al., 2018). By construction, a graphon acts as a generative model for random graphs, allowing the sampling of graphs that exhibit similar structural properties. To generate an undirected graph $\mathcal{G}$ with $N$ nodes from a given graphon $W$, the process consists of two main steps: (1) assigning each node a latent variable drawn uniformly at random from the interval $[0,1]$, and (2) connecting each pair of nodes with a probability given by evaluating $W$ at their respective latent variable values. Formally, the steps are as follows:

$$\boldsymbol{\eta}(i) \sim \text{Uniform}([0,1]), \quad \forall\, i = 1, \cdots, N, \qquad (1)$$
$$\mathbf{A}(i,j) \sim \text{Bernoulli}\left(W(\boldsymbol{\eta}(i), \boldsymbol{\eta}(j))\right), \quad \forall\, i,j = 1, \cdots, N,$$

where the latent variables $\boldsymbol{\eta}(i) \in [0,1]$ are independently drawn for each node $i$. The dissimilarity between two graphons, $W_1$ and $W_2$, is measured by the cut distance, denoted as $d_{\text{cut}}(W_1, W_2)$ (Lovász, 2012).

**Graphon estimation.**    The generative process in (1) can also be viewed in reverse: given a collection of graphs (represented by their adjacency matrix) $\mathcal{D} = \{\mathbf{A}_t\}_{t=1}^{M}$ that are sampled from an *unknown* graphon $W$, estimate $W$. Several methods have been proposed for this task (Chan & Airoldi, 2014; Airoldi et al., 2013; Xu et al., 2021; Xia et al., 2023; Azizpour et al., 2025; Ramezanpour et al., 2025). We focus on SIGL (Azizpour et al., 2025), a resolution-free method that, in addition to estimating the graphon, also *infers the latent variables* $\boldsymbol{\eta}$, making it particularly useful for model-driven augmentation. This method parameterizes the graphon using an implicit neural representation (INR) (Sitzmann et al., 2020), a neural architecture defined as $f_\phi(x,y) : [0,1]^2 \rightarrow [0,1]$ where the inputs are coordinates from $[0,1]^2$ and the output approximates the graphon value $W$ at a particular position. More details of SIGL and graphon estimation are provided in Appendix D.

**Motif densities from graphons**    A graphon $W$ provides the theoretical expectation for the density of any subgraph, also known as a motif $F$. The non-induced homomorphism density, $t(F,W)$, counts all occurrences of a motif where at least the specified edges are present, regardless of any additional edges that might exist between the motif's vertices. The non-induced density is given as follows

$$t(F,W) = \int_{[0,1]^k} \prod_{(i,j) \in \mathcal{E}_F} W(\eta_i, \eta_j) \prod_{l \in \mathcal{V}_F} d\eta_l. \quad (2)$$

The power of this framework lies in its connection to observable data; the theoretical value of $t(F,W)$ can be accurately estimated by the empirical motif counts found in a large graph sampled from the graphon $W$ (Lovász, 2012). This makes motif densities a powerful tool for linking continuous graphon models to discrete, real-world networks.

### 2.2. Graph Mixup

Mixup (Zhang et al., 2018; Verma et al., 2019) has been successfully adapted to graphs in several works (Han et al.,

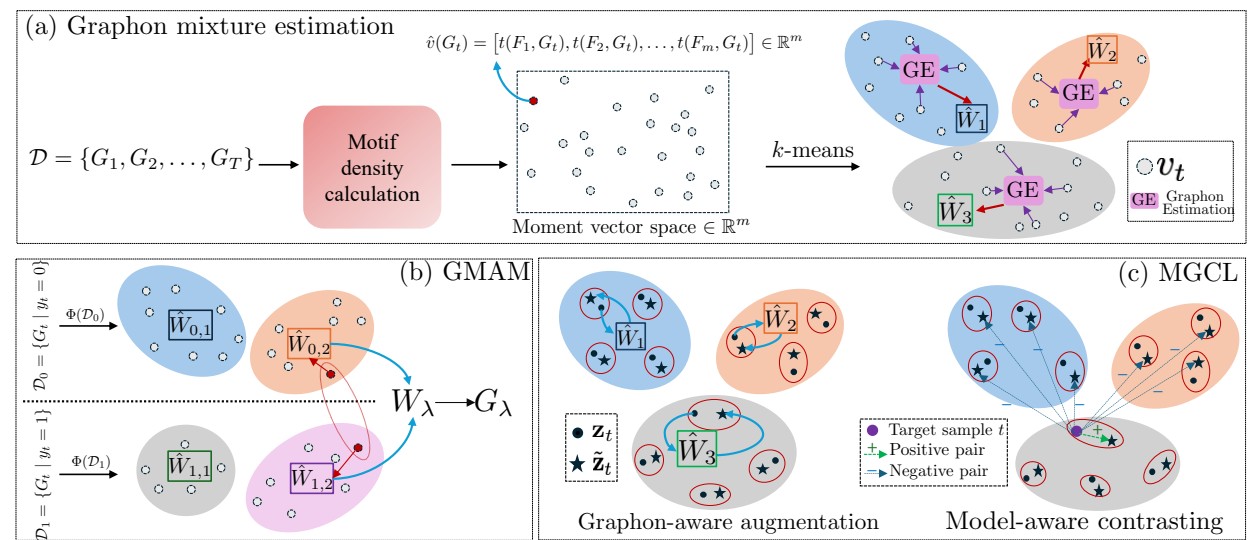

*Figure 1.* Overview of the proposed framework. (a) Graphon mixture estimation via motif moment vectors, (b) Graphon mixture–aware mixup for data augmentation, (c) Model-aware GCL leveraging graphon-informed augmentations and model-aware contrastive loss.

2022; Navarro & Segarra, 2023). The key idea is to augment the dataset by interpolating existing samples. In its standard form, mixup generates a new sample by linearly combining two randomly chosen inputs and their labels:

$$x_{\text{new}} = \lambda x_i + (1 - \lambda)x_j, \quad y_{\text{new}} = \lambda y_i + (1 - \lambda)y_j, \quad (3)$$

where $(x_i, y_i)$ and $(x_j, y_j)$ are two samples with one-hot labels. In the graph domain, G-Mixup extends this idea by operating at the level of graphons (Han et al., 2022; Navarro & Segarra, 2023). Given two classes, it first estimates their graphons, then interpolates between them, and finally generates new graphs and labels from the mixed graphon:

$$\text{Graphon estimation: } \mathcal{D}_0 \rightarrow W_{\mathcal{G}}, \quad \mathcal{D}_1 \rightarrow W_{\mathcal{H}},$$
$$\text{Graphon mixup: } W_{\mathcal{I}} = \lambda W_{\mathcal{G}} + (1 - \lambda)W_{\mathcal{H}},$$
$$\text{Graph generation: } \{I_1, \ldots, I_m\} \overset{\text{i.i.d.}}{\sim} \mathbb{G}(K, W_{\mathcal{I}}),$$
$$\text{Label mixup: } \mathbf{y}_{\mathcal{I}} = \lambda \mathbf{y}_{\mathcal{G}} + (1 - \lambda)\mathbf{y}_{\mathcal{H}}.$$

### 2.3. Graph contrastive learning

GCL (Veličković et al., 2019; You et al., 2020; 2021; Suresh et al., 2021) aims to learn discriminative node or graph-level embeddings in a self-supervised manner, without relying on explicit labels. Given a collection of graphs $\{\mathcal{G}_t\}_{t=1}^T$, the goal of graph-level GCL is to train an encoder $\mathcal{E}_\theta(\cdot)$ so that it produces expressive representations or embeddings. The encoder outputs an embedding per graph $\mathcal{G}_t$, denoted by $\mathbf{z}_t = \mathcal{E}_\theta(\mathbf{A}_t, \mathbf{X}_t)$, where $\mathbf{z}_t \in \mathbb{R}^F$. InfoNCE-based methods are widely adopted in this setting (You et al., 2020; Suresh et al., 2021; You et al., 2021). These methods generate an augmented view of the input graph through transformations such as edge perturbation, feature masking, or subgraph sampling. Following this, let $\mathbf{z}_t$ and $\tilde{\mathbf{z}}_t$

denote the graph-level representations of the original and augmented views, respectively. The InfoNCE loss is then used to bring $\mathbf{z}_t$ and $\tilde{\mathbf{z}}_t$ (positive pair) closer in the embedding space while pushing them apart from representations of all other graphs in the batch ($\tilde{\mathbf{z}}_{t'}$, for all $t' \neq t$, i.e., negative samples). Formally, the parameters of the encoder are achieved by minimizing the following loss function

$$\mathcal{L}_t = -\frac{1}{L}\sum_{t=1}^L \log \frac{\exp(\text{sim}(\mathbf{z}_t, \tilde{\mathbf{z}}_t)/\tau)}{\sum_{t'=1,\, t' \neq t}^L \exp(\text{sim}(\mathbf{z}_t, \tilde{\mathbf{z}}_{t'})/\tau)}. \quad (4)$$

## 3. Methods

We first present our unified framework for estimating the multiple underlying data distributions of graphs as a graphon mixture in Section 3.1. We then introduce how to adapt and leverage these estimated graphons in the context of graph mixup and graph contrastive learning, as detailed in Sections 3.2 and 3.3, respectively. An overview of these steps is shown in Figure 1, and a detailed algorithm of these three steps is included in Appendix A.

### 3.1. Graphon Mixture Estimation

The goal of this step, which is the core of our methodology, is to recover the multiple underlying generative models of the data, represented as a set of graphons $\{W_i\}_{i=1}^K$. We assume that each observed graph $G_t$ in the dataset is sampled from one of these graphons, though the assignment is unknown. Hence, the task reduces to identifying groups of graphs that are likely to originate from the same underlying distribution, and subsequently estimating a representative graphon for each group.

To formalize this idea, we leverage the concept of *mo-*

*ment vectors.* For a graphon $W$, its motif densities provide expectations of subgraph counts (Equation (2)). Let $\mathcal{F} = \{F_1, \ldots, F_m\}$ denote a fixed family of motifs. The *moment vector* of a graphon $W$ with respect to $\mathcal{F}$ is

$$v(W) = \big[t(F_1, W), t(F_2, W), \ldots, t(F_m, W)\big] \in \mathbb{R}^m. \tag{5}$$

Similarly, for an observed graph $G$ sampled from $W$, we compute its empirical moment vector $\hat{v}(G)$ by normalizing motif counts in $G$. We present next a result to bound the difference between the empirical moment vectors of two graphs.

**Theorem 3.1** (Bounding Empirical Density Difference by Graphon Distance). *Let $G_1$ and $G_2$ be $n$-vertex graphs sampled from graphons $W_1$ and $W_2$, respectively, where $d_{cut}(W_1, W_2) \leq \epsilon$. Let $\mathcal{F}$ be a fixed family of motifs of bounded size. For any motif $F \in \mathcal{F}$ with $k = v(F)$ vertices and $e(F)$ edges, and for any target failure probability $\eta > 0$, we have*

$$|t(F, G_1) - t(F, G_2)| \leq \underbrace{e(F)\epsilon}_{\text{Graphon difference}}$$

$$+ 2\underbrace{\left(\sqrt{\frac{1}{2m}\log\frac{4}{\eta}} + \frac{e(F)}{\sqrt{n(n-1)}}\sqrt{2\log\frac{4}{\eta}}\right)}_{\text{Sampling error}}. \tag{6}$$

*with probability at least $1 - 2\eta$, where $m = \lfloor\frac{n}{k}\rfloor$. The proof is provided in Appendix B.*

The bound in Equation (6) on the total difference in empirical motif densities is comprised of two components. The first term, the *graphon difference*, arises from the intrinsic structural dissimilarity between the underlying graphons $W_1$ and $W_2$. The second term, the *sampling error*, represents a novel and tighter concentration bound that quantifies the statistical uncertainty from the random graph generation process. This novel bound offers a significant improvement over classical approaches that can be derived from the literature (Lovász, 2012; Borgs et al., 2008), as we demonstrate in Appendix B.1, where a full derivation and comparison are provided.

Theorem 3.1 reveals that if the underlying graphons $W_1$ and $W_2$ are close in cut distance, i.e., small $\epsilon$, then the corresponding empirical moment vectors, $\hat{v}(G_1)$ and $\hat{v}(G_2)$, are also close with high probability. Alternatively, when the sampling error is sufficiently small (e.g., for sufficiently large graphs), a large observed discrepancy between the empirical moment vectors of two graphs provides evidence that their underlying graphons are not close in cut distance. Formally, in this small-sampling-error regime, if $|t(F, G_1) - t(F, G_2)|$ is large for some motif $F$, it implies that $d_{cut}(W_1, W_2)$ must also be large. By applying a union bound across all motifs in the family $\mathcal{F}$, this principle extends to the full moment vector: a large Euclidean distance

$\|\hat{v}(G_1) - \hat{v}(G_2)\|_2$ implies that $G_1$ and $G_2$ likely originate from different graphon distributions. This justifies the use of motif-based embeddings for graph clustering, as distinct generating processes produce measurably distinct motif fingerprints. Further theoretical analyses are included in Appendix B.1.

Equipped with the above theorem, we assign each graph $G_t$ a feature vector $v_t = \hat{v}(G_t) \in \mathbb{R}^m$, and collect them as $\{v_t\}_{t=1}^T$. We then apply $k$-means clustering to these vectors in order to group graphs according to their latent generative models. Let $\{C_1, \ldots, C_K\}$ denote the resulting clusters. Within each cluster $C_k$, the graphs share similar moment vectors, suggesting they arise from a common graphon $W_k$. Then, we aim to estimate a graphon for each cluster of graphs. To mitigate boundary effects (i.e., graphs near the edges of clusters that may not be well represented), we select the $L$ graphs closest to the cluster centroid in moment space. For each cluster $C_k$, we estimate the underlying graphon $W_k$ using these $L$ graphs. Any graphon estimation algorithm can be applied; in our implementation, we adopt SIGL (Azizpour et al., 2025), which additionally provides the latent node variables $\boldsymbol{\eta}$ alongside the graphon estimate. Finally, each graph $G_t$ in cluster $C_k$ is associated with the estimated graphon $W_k$. This provides us with a graphon mixture model $\{W_1, \ldots, W_K\}$, where each observed graph is assigned to exactly one mixture component. To align with the dataset size, we set $K = \log T$ as a simple, dataset-size-aware default. Since $K$ acts on the motif-embedding space, it can equivalently be selected using standard cluster-number heuristics (e.g., the elbow method or silhouette analysis); we leave $K$ as a tunable hyperparameter and study its effect empirically in Appendix F.4.

In summary, we can view the overall pipeline as a mapping from a collection of graphs to a set of estimated graphons together with an assignment function. Let $\mathcal{D} = \{G_1, G_2, \ldots, G_T\}$ denote the dataset of observed graphs. Our procedure outputs (i) a set of estimated graphons $\{\hat{W}_1, \ldots, \hat{W}_K\}$ and (ii) an assignment function

$$\pi : \mathcal{D} \to \{1, \ldots, K\},$$

such that each graph $G_t$ is mapped to one of the estimated graphons $\hat{W}_{\pi(G_t)}$. Equivalently, the pipeline defines a function

$$\Phi : \mathcal{D} \longrightarrow \big(\{\hat{W}_1, \ldots, \hat{W}_K\}, \pi\big), \tag{7}$$

where $\Phi$ encapsulates the steps of (i) computing empirical moment vectors, (ii) clustering in moment space, and (iii) estimating graphons for each cluster. A detailed computational complexity analysis of the graphon mixture estimation procedure $\Phi$ is provided in Appendix A.1.

## 3.2. Graphon Mixture-Aware Mixup (GMAM)

In this section, we describe how to leverage the graphon mixture for graph data augmentation under the mixup framework. This serves as a concrete example of how incorporating underlying generative models can benefit supervised downstream learning. In contrast to existing G-mixup, which assumes a single graphon per class, our approach disentangles the multiple generative models that may exist within each class. This enables a more fine-grained and semantically valid interpolation strategy.

Consider a dataset of graphs with $C$ classes, $\mathcal{D} = \{(G_t, y_t)\}_{t=1}^{T}$, where $y_t \in \{1, \ldots, C\}$. For each class $i$, we first collect its subset of graphs $\mathcal{D}_i = \{G_t \mid y_t = i\}$. We then apply the operator $\Phi$ from Section 3.1 to estimate the graphon mixture within each class. Formally, this yields

$$\Phi(\mathcal{D}_i) \longrightarrow \left(\{\hat{W}_{i,1}, \ldots, \hat{W}_{i,K_i}\}, \pi_i\right), \quad (8)$$

where $\{\hat{W}_{i,1}, \ldots, \hat{W}_{i,K_i}\}$ denotes the mixture of graphons estimated for class $i$, and $\pi_i$ is the assignment of each graph in $\mathcal{D}_i$ to its underlying graphon. To perform GMAM, we first sample two graphs $G_a \in \mathcal{D}_i$ and $G_b \in \mathcal{D}_j$ from classes $i \neq j$. Instead of directly interpolating their raw structure, we trace back to their associated graphons, $\hat{W}_{i,\pi_i(G_a)}$ and $\hat{W}_{j,\pi_j(G_b)}$. We then construct a mixed graphon

$$W_\lambda = \lambda \hat{W}_{i,\pi_i(G_a)} + (1 - \lambda)\hat{W}_{j,\pi_j(G_b)}. \quad (9)$$

Finally, we generate a new graph $G_\lambda \sim \mathbb{G}(n, W_\lambda)$ according to the stochastic sampling process in (1), where $n$ is chosen to match the scale of the dataset. The associated label is interpolated in the standard mixup fashion: $\mathbf{y}_\lambda = \lambda \mathbf{y}_i + (1 - \lambda)\mathbf{y}_j$, where $\mathbf{y}_i, \mathbf{y}_j$ are one-hot encoded labels. By disentangling class distributions into mixtures of graphons, we preserve finer semantic structures during augmentation and avoid the unrealistic assumption that all graphs within a class share a single generative model. This leads to higher-quality data augmentation and, consequently, improved graph classification performance. We evaluate this effect in Section 4.2.

## 3.3. Model-aware Graph Contrastive Learning (MGCL)

We now extend the graphon mixture framework to contrastive learning, providing a second example of how incorporating underlying generative models can improve downstream learning in an unsupervised setting. Given an unlabeled dataset of graphs $\mathcal{D} = \{G_t\}_{t=1}^{T}$, we first apply the operator $\Phi$ (Section 3.1) to obtain the graphon mixture $\{\hat{W}_1, \ldots, \hat{W}_K\}$ along with the assignment $\pi$ that maps each graph $G_t$ to its generating graphon $\hat{W}_{\pi(G_t)}$. This mapping benefits contrastive learning in two key ways: (1) it enables a principled graphon-aware augmentation strategy that generates meaningful positive pairs, and (2) it supports a model-aware loss function.

**Graphon-aware augmentation.** For each cluster $k$, the estimated graphon $\hat{W}_k$ characterizes the generative distribution of graphs in that cluster. We use this distribution to design a graphon-informed augmentation procedure. Given a graph $G_t$ with adjacency matrix $\mathbf{A}_t$ that belongs to cluster $C_t$, we randomly select a subset $E_{\text{sel}}$ of $r\%$ of all node pairs. For each selected pair $(i, j) \in E_{\text{sel}}$, the corresponding adjacency entry is resampled according to the probability given by the cluster's graphon:

$$\tilde{\mathbf{A}}_t(i, j) \sim \text{Bernoulli}\left(\hat{W}_{C_t}(\eta_i, \eta_j)\right), \tilde{\mathbf{A}}_t(j, i) = \tilde{\mathbf{A}}_t(i, j), \quad (10)$$

where $\eta_i, \eta_j \in [0, 1]$ denote the latent positions associated with nodes $i$ and $j$. For all pairs $(i, j) \notin E_{\text{sel}}$, we retain the original edges: $\tilde{\mathbf{A}}_t(i, j) = \mathbf{A}_t(i, j)$. Note that, to leverage graphons for augmentation and edge resampling, we require the latent variables of the nodes ($\eta$) in order to query the edge probabilities from the graphon. Since SIGL (Azizpour et al., 2025) is specifically designed to estimate graphons together with the latent node positions, we adopt it for graphon estimation. Nevertheless, other graphon estimation methods (Chatterjee, 2015; Chan & Airoldi, 2014) can also be employed, as they implicitly assign latent variables by sorting nodes according to degree and using the degree as a proxy for the latent position.

This process injects structure-aware perturbations guided by the estimated generative model. Unlike naive random perturbations-such as edge or node dropping-commonly used in existing graph contrastive learning methods, graphon-aware augmentation assigns edge-specific probabilities informed by the graphon, producing more faithful augmented views. Passing the original graph and its graphon-aware augmentation through the encoder $\mathcal{E}_\theta$ with node features $\mathbf{X}_t$ yields the corresponding representations

$$\begin{aligned} \mathbf{z}_t &= \mathcal{E}_\theta(\mathbf{A}_t, \mathbf{X}_t), \\ \tilde{\mathbf{z}}_t &= \mathcal{E}_\theta(\tilde{\mathbf{A}}_t, \mathbf{X}_t). \end{aligned} \quad (11)$$

Moreover, prior work has established a theoretical connection between graphons and the stability of graph neural network representations. In particular, Ruiz et al. (2020) shows that when two graphs are generated from the same underlying graphon, the distance between their GNN outputs (($\mathbf{z}_t$ and $\tilde{\mathbf{z}}_t$ in our setting) is bounded, with the bound depending on properties of the graphon and the graph resolutions. This provides a principled justification for graphon-aware augmentation: by generating augmented views according to the same underlying graphon, the resulting representations are guaranteed to remain close in the embedding space. In contrast, commonly used random perturbations in existing graph contrastive learning frameworks offer no such guarantee and may produce augmented graphs that deviate substantially from the anchor's generative model. Consequently, conditioning augmentations on the inferred graphon

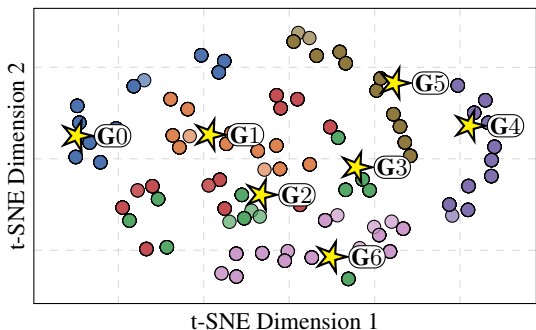 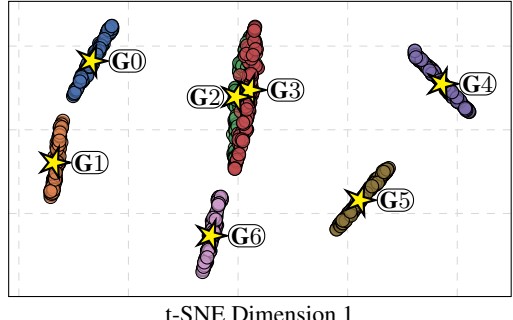

*Figure 2.* t-SNE visualization of empirical moment vectors for graphs sampled from a mixture of graphons. Left: *Varying* size, $n \sim U[75, 300]$. Right: *Fixed* size, $n = 200$. Each color represents a different graphon.

leads to more meaningful positive pairs and better-aligned contrastive objectives.

**Model-aware contrasting.** We modify the InfoNCE objective (Oord et al., 2018) to account for graphon mixture assignments. Given a batch of $L$ graphs, we define a new loss for anchor $t$, and the overall loss function is

$$\mathcal{L}_{\text{all}} = -\frac{1}{L} \sum_{t=1}^{L} \log \frac{\exp(\text{sim}(\mathbf{z}_t, \tilde{\mathbf{z}}_t)/\tau)}{\sum\limits_{t'=1, \, C_{t'} \neq C_t}^{L} \exp(\text{sim}(\mathbf{z}_t, \tilde{\mathbf{z}}_{t'})/\tau)},$$
(12)

where $\text{sim}(\cdot, \cdot)$ is a similarity measure (e.g., cosine), $\tau$ is the temperature, and $C_t$ is the cluster index of $G_t$. Unlike standard InfoNCE, which pushes a graph away from all other graphs (including those with the same underlying model), our formulation only contrasts against graphs from *different* graphons. The following result formalizes how the proposed loss enforces model-aware representations through its contrastive structure.

**Theorem 3.2** (Lower bound of model-aware loss). *For every graph $t$ let $m_t := |\{k : C_k \neq C_t\}|$, and $\bar{\mathbf{z}}^{(\neg C_t)} := \frac{1}{m_t} \sum_{C_k \neq C_t} \tilde{\mathbf{z}}_k$ be the centroid of negative samples from clusters other than $C_t$. Then,*

$$\ln m_t + \text{sim}(\mathbf{z}_t, \bar{\mathbf{z}}^{(\neg C_t)}) - \text{sim}(\mathbf{z}_t, \tilde{\mathbf{z}}_t) \leq \ell_t, \quad (13)$$

*where $\ell_t = \log \frac{\exp(\text{sim}(\mathbf{z}_t, \tilde{\mathbf{z}}_t)/\tau)}{\sum\limits_{t'=1, C_{t'} \neq C_t}^{L} \exp(\text{sim}(\mathbf{z}_t, \tilde{\mathbf{z}}_{t'})/\tau)}$, is the loss of anchor $t$.*

The proof is provided in Appendix C. This lower bound shows that minimizing the proposed loss encourages each graph representation $\mathbf{z}_t$ to separate from the centroid of representations corresponding to graphs generated by *different* underlying models, as captured by the term $\text{sim}(\mathbf{z}_t, \bar{\mathbf{z}}^{(\neg C_t)})$. In contrast to standard contrastive learning, where all other graphs in the batch are treated as negatives, our formulation restricts negative samples to graphs drawn from different graphons, thereby excluding structurally similar graphs

from the same mixture component and reducing false negatives (see Appendix F.6). Moreover, the use of graphon-aware augmentations ensures that the positive sample $\tilde{\mathbf{z}}_t$ remains semantically consistent with the underlying generative process, strengthening the alignment term $\text{sim}(\mathbf{z}_t, \tilde{\mathbf{z}}_t)$. Together, these effects encourage the encoder to align each graph representation with its generating model while explicitly separating it from other models. As a result, the proposed objective can be interpreted as performing contrast at the *model level*, contrasting graphs against representations of negative models rather than individual instances, yielding representations that are more discriminative and semantically faithful.

## 4. Experiments

We evaluate our proposed pipeline along three main directions. First, in Section 4.1, we test whether clustering based on moment vectors can successfully separate graphs generated from different underlying models using a variety of simulated settings. Next, in Sections 4.2 and 4.3, we assess the effectiveness of mixture-aware extensions on real-world downstream tasks, namely GMAM and MGCL, respectively. Provided in Appendix E for all three types of experiments are details on hyperparameter settings (Appendix E.2), the datasets (Appendix E.3), baseline methods (Appendix E.4), and computational resources (Appendix E.5); our code is available at https://github.com/aliaaz99/moments2models.

### 4.1. Synthetic experiment

To empirically validate the theoretical framework presented in Section 3.1, we conduct a synthetic experiment designed to test the efficacy of moment vectors in distinguishing and clustering graphs sampled from a mixture of different graphon distributions. The core hypothesis is that graphs generated from distinct graphons will exhibit measurably different motif densities, forming separable clusters in the moment vector space. We generate a dataset composed of graphs sampled from a mixture of $K = 7$ distinct, prede-

*Table 1.* Graph classification accuracy of different mixup methods compared with GMAM. (Notation: **Best**, second best)

| Methods | IMDB-B | IMDB-M | REDD-B | REDD-M5 | REDD-M12 | COLLAB | AIDS |
|---|---|---|---|---|---|---|---|
| Vanilla | $71.30 \pm 4.36$ | $48.80 \pm 2.54$ | $89.15 \pm 2.47$ | $53.17 \pm 2.26$ | $49.95$ $\pm 0.98$ | $79.39 \pm 1.24$ | $98.00 \pm 1.20$ |
| DropEdge | $70.50 \pm 3.80$ | $48.73 \pm 4.08$ | $87.45 \pm 3.91$ | $54.11 \pm 1.94$ | $49.77 \pm 0.76$ | $78.32 \pm 1.31$ | $92.87 \pm 1.45$ |
| DropNode | $72.00 \pm 6.97$ | $45.67 \pm 2.59$ | $88.60 \pm 2.52$ | $53.97 \pm 2.11$ | $49.95$ $\pm 1.70$ | $80.01 \pm 1.66$ | $96.25 \pm 1.24$ |
| SubMix | $71.70 \pm 6.20$ | $49.80 \pm 4.01$ | $90.45 \pm 1.93$ | $54.27 \pm 2.92$ | $42.58 \pm 12.30$ | $80.11 \pm 0.75$ | $96.18 \pm 1.33$ |
| M-Mixup | $72.00 \pm 5.14$ | $48.67 \pm 5.32$ | $87.70 \pm 2.50$ | $52.85 \pm 1.03$ | $49.81 \pm 0.80$ | $77.00 \pm 2.20$ | $-$ |
| S-Mixup | $73.40 \pm 6.26$ | $50.13 \pm 4.34$ | $90.55 \pm 2.11$ | $55.19 \pm 1.99$ | $49.51 \pm 1.59$ | $-$ | $-$ |
| *G*-Mixup | $72.40 \pm 5.64$ | $49.93 \pm 2.82$ | $90.20 \pm 2.84$ | $54.33 \pm 1.99$ | $49.72 \pm 0.48$ | $79.05 \pm 1.25$ | $97.80 \pm 0.90$ |
| SIGL | $73.95 \pm 2.64$ | $50.70 \pm 1.41$ | $91.93$ $\pm 0.94$ | $55.82 \pm 1.35$ | $49.73 \pm 0.62$ | $80.15$ $\pm 0.60$ | $97.93 \pm 0.96$ |
| MomentMixup | $74.30$ $\pm 2.70$ | $50.95$ $\pm 1.93$ | $91.80 \pm 1.20$ | $56.09$ $\pm 1.62$ | $49.83 \pm 1.01$ | $79.75 \pm 0.59$ | **$98.50$** $\pm 0.60$ |
| GMAM | **$74.45$** $\pm 1.15$ | **$51.03$** $\pm 1.63$ | **$92.25$** $\pm 0.82$ | **$56.46$** $\pm 0.95$ | **$50.18$** $\pm 0.50$ | **$80.25$** $\pm 0.52$ | $98.20$ $\pm 0.51$ |

fined graphon models, $\{W_k\}_{k=0}^6$ (Appendix E.1). These models, include a range of functional forms to ensure structural diversity, generating graphs of varying density through models that are, for example, linear or exponential in nature. For each graphon model $W_k$, we sample a collection of graphs, creating a dataset where the ground-truth generating model for each graph is known.

To analyze the impact of graph size on the concentration of empirical moments around their theoretical means (Theorem 3.1), we structure our experiment into two distinct scenarios: (i) *Varying* size, where $n$ is drawn uniformly from $[75, 300]$; and (ii) *Fixed* size, with $n = 200$. For each scenario, we generate a balanced dataset with an equal number of graphs from each of the seven graphon classes. For each sampled graph, we compute its empirical moment vector using the densities of all connected motifs with up to 4 nodes, resulting in feature vectors of size 9. We then apply the clustering algorithm introduced in Section 3.1 to cluster these vectors and measure the accuracy by comparing the assignments to the ground-truth labels. This directly tests the ability of moment vectors to partition a mixture of graphs generated from different underlying models. We visualize a low-dimensional representation of the moment vectors by projecting them into two dimensions using t-SNE in Figure 2.

In these plots, each point represents a graph sampled from one of seven graphon distributions, and the stars denote the ground-truth moment vectors $v(W_k)$ for each underlying graphon $W_k$. The visualizations empirically confirm our theoretical framework. According to Theorem 3.1, the distance between the empirical moment vectors of two sampled graphs, $\hat{v}(G_1)$ and $\hat{v}(G_2)$, is influenced by two primary factors, namely the distance between their underlying graphons, $d_{\text{cut}}(W_1, W_2)$, and a sampling error term which decreases as the number of nodes $n$ grows. In the *varying* size scenario (Figure 2(a)), the higher variance in graph sizes leads to more diffuse clusters, as smaller graphs introduce larger sampling errors. Conversely, in the *fixed* size scenario with $n = 200$ (Figure 2(b)), the empirical vectors concentrate more tightly around their respective ground-truth means, forming more distinct and separable clusters. The separa-

tion between clusters is determined by the distance between their underlying graphons. Table 5 (Appendix E.1) reports the pairwise Gromov-Wasserstein (GW) distances, used here as a practical proxy for the cut distance since the latter can be relaxed to the GW distance of step functions (Xu et al., 2021). As expected, graphons 2 and 3 have a very small GW distance of $0.024$, reflected in the overlap of their green and red clusters, while graphons with larger distances, such as 0 and 4 ($0.530$), yield well-separated clusters.

*Table 2.* Clustering accuracy (%) with different embeddings.

| Method | Varying | Fixed |
|---|---|---|
| Theory | 81.4 | 82.9 |
| GCN | 58.6 | 64.7 |
| GIN | 25.7 | 60.9 |
| Graph2Vec | 28.6 | 62.0 |
| DeepWalk | 25.7 | 28.9 |
| Spectral | 22.9 | 21.7 |
| DisenGCN | 18.0 | 19.2 |
| MBC (our) | **80.0** | **79.3** |

Quantitatively, our clustering performance is reported in Table 2. The Theory-Based accuracy is computed by assigning each graph to the cluster of the nearest ground-truth moment vector in Euclidean space, providing an upper bound on performance. Other methods apply k-means clustering on the embeddings obtained by different methods. Our proposed method, MBC (moment-based clustering), achieves accuracies of $80.0$ and $79.3$ for the varying and fixed size settings, respectively, outperforming all other embedding methods and closely approaching the theory-based scores of $81.4$ and $82.9$. This explicitly demonstrates that moment vectors provide a robust foundation for clustering graphs generated from different underlying graphons. The higher accuracy in the fixed-size setting directly reflects the tighter cluster formations observed in the t-SNE visualization, further validating the concentration properties outlined in our theory.

Further experimental results are reported in the appendix, including an ablation study on the number of motifs (Appendix F.1), the effect of graph size on moment concentration (Appendix F.2), an analysis of the effect of the number of clusters (Appendix F.4.1), an investigation of varying the

number of mixture components (Appendix F.5), and evaluating the quality of the estimated graphons (Appendix F.3).

## 4.2. GMAM

In this section, we evaluate the effectiveness of our proposed mixture-aware graph mixup method in Section 3.2 on seven real-world datasets from the TUDatasets benchmark (Morris et al., 2020), which are commonly used for mixup evaluation. We compare against a vanilla baseline without augmentation, as well as a broad range of simple augmentations and state-of-the-art mixup variants. Full details on datasets and baselines are provided in Appendix E.3 and Appendix E.4. For fairness, we adopt the same GNN structure (depth and architecture) across all methods, along with identical training hyperparameters. All steps of graphon mixture estimation and the subsequent mixup augmentation are performed strictly within the training set. The augmentation ratio (i.e., the proportion of generated graphs added to training) is also fixed consistently across methods.

Table 1 reports the results. Our method consistently improves performance across multiple datasets, achieving the highest accuracy on six of seven benchmarks and ranking second on the remaining one. Notably, when restricting our framework to a single graphon per class, GMAM reduces to the SIGL-based mixup variant reported in Table 1, which GMAM consistently outperforms. These results demonstrate that explicitly modeling class distributions as mixtures of graphons rather than assuming a single graphon per class leads to higher-quality augmentations and improved graph classification performance, outperforming existing augmentation strategies, including those that do not rely on graphon-based mixup. An ablation study examining the effect of the number of estimated mixture components is presented in Appendix F.4.2. In addition, the estimated underlying graphons for different classes are illustrated in Appendix F.8, revealing the presence of distinct generative models across classes.

## 4.3. MGCL

We compare MGCL with a variety of state-of-the-art GCL baselines. The full set of competing methods is detailed in Appendix E.4. We evaluate on seven datasets from the TUDataset (Morris et al., 2020) collection. The experiments follow the linear evaluation protocol (Peng et al., 2020) in the literature and previous works, where models are first trained in an unsupervised manner, and the resulting embeddings are subsequently used for downstream tasks. We adopt the evaluation protocol from (Sun et al., 2020), performing 10-fold cross-validation on each dataset. The resulting graph-level embeddings are used to train an SVM classifier, and we report the average performance across the folds. To summarize performance, we assign a rank to each

method based on its performance on each dataset, and the average rank (A.R.) is then computed as the mean of these ranks across all datasets, consistent with prior work.

As reported in Table 3, MGCL demonstrates strong performance across a broad range of graph-level benchmarks. MGCL ranks first on four out of seven datasets and second on the remaining three. On IMDB-B, InfoGraph attains the best performance, suggesting that augmentation-based strategies may be less effective for this particular benchmark. Nevertheless, MGCL consistently outperforms other augmentation-based methods, such as GraphCL and JOAO, as well as more recent approaches including DRGCL and simGRACE. Overall, MGCL achieves the lowest average rank (A.R.) of 1.42 among the competing methods, indicating its ability to learn robust and transferable graph-level representations. Importantly, MGCL constitutes a simple yet principled extension of GraphCL: by explicitly modeling the dataset as a mixture of underlying generative models and replacing random augmentations with graphon-aware augmentations, it consistently improves upon GraphCL and achieves performance comparable to, or better than, more complex recent methods that either learn augmentations or introduce sophisticated contrastive objectives. These results suggest that incorporating graphon mixture estimation into a standard contrastive learning pipeline can lead to meaningful performance improvements.

Additional analyses further support the effectiveness of the proposed model-aware graph contrastive learning framework. In particular, model-aware clustering is shown to reduce the false negative rate in contrastive learning (Appendix F.6). An ablation study on the number of clusters used in MGCL is provided in Appendix F.4.3, and the estimated graphons are illustrated in Figure 9 in Appendix F.8. We empirically disentangle the contributions of graphon-informed augmentation (GIA) and cluster-aware loss (CAL) in Appendix F.7. Finally, a detailed runtime comparison of incorporating the graphon mixture estimation step into both Mixup and graph contrastive learning pipelines is provided in Appendix A.2.

## 5. Conclusion

We proposed a unified framework for inferring mixtures of underlying generative models (graphons) from observed graphs. Instead of assuming a single distribution per dataset, our approach disentangles multiple generative models and leverages this structure for downstream learning. To the best of our knowledge, we introduced the first method that estimates a graphon mixture from observed graphs based on motif moment vectors and uses it for downstream graph learning, going beyond prior single-graphon approaches. This is supported by tighter concentration guarantees for graph moments, and we incorporated the estimated mix-

*Table 3.* Unsupervised representation learning classification accuracy (%). A.R. denotes the average rank. (Notation: **Best**, second best)

| Methods | NCI1 | PROTEINS | DD | MUTAG | COLLAB | RDT-B | IMDB-B | A.R. |
|---|---|---|---|---|---|---|---|---|
| InfoGraph | $76.20 \pm 1.0$ | $74.44 \pm 0.3$ | $72.85 \pm 1.8$ | $89.01 \pm 1.1$ | $70.65 \pm 1.1$ | $82.50 \pm 1.4$ | **73.03** $\pm 0.9$ | 7.00 |
| GraphCL | $77.87 \pm 0.4$ | $74.39 \pm 0.4$ | $78.62 \pm 0.4$ | $86.80 \pm 1.3$ | $71.36 \pm 1.1$ | $89.53 \pm 0.8$ | $71.14 \pm 0.4$ | 6.42 |
| MVGRL | $76.64 \pm 0.3$ | $74.02 \pm 0.3$ | $75.20 \pm 0.4$ | $75.40 \pm 7.8$ | **73.10** $\pm 0.6$ | $82.00 \pm 1.10$ | $63.60 \pm 4.2$ | 8.85 |
| JOAO | $78.07 \pm 0.5$ | $74.55 \pm 0.4$ | $77.32 \pm 0.5$ | $87.35 \pm 1.0$ | $69.50 \pm 0.3$ | $85.29 \pm 1.3$ | $70.21 \pm 3.1$ | 7.85 |
| JOAOv2 | $78.36 \pm 0.5$ | $74.07 \pm 1.1$ | $77.40 \pm 1.1$ | $87.67 \pm 0.8$ | $69.33 \pm 0.3$ | $86.42 \pm 1.4$ | $70.83 \pm 0.3$ | 7.28 |
| AD-GCL | $73.90 \pm 0.4$ | $73.30 \pm 0.5$ | $75.80 \pm 0.9$ | $88.70 \pm 1.1$ | $72.00 \pm 0.6$ | $90.10 \pm 0.9$ | $70.20 \pm 0.7$ | 7.28 |
| AutoGCL | $78.32 \pm 0.5$ | $69.73 \pm 0.4$ | $75.75 \pm 0.6$ | $85.15 \pm 1.1$ | $71.40 \pm 0.7$ | $86.60 \pm 1.5$ | $72.00 \pm 0.4$ | 7.14 |
| simGRACE | $79.10 \pm 0.4$ | $75.30 \pm 0.1$ | $77.40 \pm 1.1$ | $89.00 \pm 1.3$ | $71.70 \pm 0.8$ | $89.50 \pm 0.9$ | $71.30 \pm 0.8$ | 4.14 |
| RGCL | $78.10 \pm 1.0$ | $75.00 \pm 0.4$ | $78.90 \pm 0.5$ | $87.70 \pm 1.0$ | $71.00 \pm 0.7$ | $90.30 \pm 0.6$ | $71.90 \pm 0.9$ | 4.71 |
| DRGCL | $78.70 \pm 0.4$ | $75.20 \pm 0.6$ | $78.40 \pm 0.7$ | $89.50 \pm 0.6$ | $70.60 \pm 0.8$ | **90.80** $\pm 0.3$ | $72.00 \pm 0.5$ | 3.57 |
| MGCL | **79.18** $\pm 0.48$ | **75.54** $\pm 0.31$ | **79.07** $\pm 0.24$ | **90.03** $\pm 1.32$ | $72.28 \pm 0.63$ | $90.46 \pm 0.14$ | $72.28 \pm 0.52$ | **1.42** |

ture structure into two widely studied downstream graph learning tasks. Specifically, we developed graphon-mixture-aware mixup (GMAM) for model-level data augmentation in supervised settings, and model-aware graph contrastive learning (MGCL), which combines graphon-informed augmentations with a principled, model-aware loss for unsupervised representation learning. Empirically, we showed that moment-based clustering recovers ground-truth models, GMAM achieves state-of-the-art results on six benchmarks, and MGCL yields more semantically faithful representations, resulting in higher accuracy in downstream task classification.

While our framework is effective, its performance may be influenced by factors such as graph size and graph sparsity. For smaller graphs in particular, higher sampling variance in motif counts can cause their embeddings to deviate from their theoretical means, making it challenging to reliably identify their true underlying generative distribution. A discussion of both the small-graph regime and the dense-graphon scope, along with possible relaxations to sparse-graph settings, is provided in Appendix G. Finally, the strong separability observed in motif-based embeddings suggests a promising direction for graph representation learning and graph-level anomaly detection, where graphs that deviate in moment space may be identified as arising from anomalous generative processes.

## Impact Statement

In this paper, we introduce a principled framework for modeling graph datasets as mixtures of underlying generative models using graph moments and graphons, along with model-aware extensions of mixup and graph contrastive learning. These contributions promote more faithful and effective graph representation learning by explicitly accounting for heterogeneity in graph-generating processes, rather than assuming a single latent distribution. The proposed framework may enable practitioners to better understand, analyze, and represent structural variability in real-world graph datasets.

## Acknowledgments

This work was supported by the NSF under grants EF-2126387 and CCF-2340481.

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

# A. Algorithms

We present the three core algorithms that constitute our framework. The first, Algorithm 1, details the operator $\Phi$ for estimating a graphon mixture model from a graph dataset by clustering graphs based on their motif densities. The subsequent algorithms leverage this estimated mixture for downstream tasks: Algorithm 2 introduces a novel mixup-based data augmentation strategy (GMAM) that interpolates between the generative graphon models, while Algorithm 3 presents a model-aware contrastive learning method (MGCL) that uses the mixture for both principled data augmentation and a more effective negative sampling strategy.

---

**Algorithm 1:** Graphon Mixture Estimation (Operator $\Phi$)

**Input** : Graph dataset $\mathcal{D} = \{G_t\}_{t=1}^T$; motif family $\mathcal{F} = \{F_1, \ldots, F_m\}$; #clusters $K = \log T$; per-cluster refinement size $L$; graphon estimator (e.g., SIGL).

**Output** : Estimated graphons $\{\hat{W}_1, \ldots, \hat{W}_K\}$; assignment $\pi : \mathcal{D} \to \{1, \ldots, K\}$.

1 **for** $t = 1$ **to** $T$ **do**
2     Compute empirical motif densities $\hat{v}(G_t) \in \mathbb{R}^m$ over $\mathcal{F}$.
3 Let $V \leftarrow [\hat{v}(G_1), \ldots, \hat{v}(G_T)]^\top \in \mathbb{R}^{T \times m}$.
4 Run $k$-means on $V$ with $K$ clusters to obtain clusters $\{C_1, \ldots, C_K\}$ and centroids $\{\mu_1, \ldots, \mu_K\}$.
5 Define assignment $\pi(G_t) \leftarrow \arg\min_{k \in \{1, \ldots, K\}} \|\hat{v}(G_t) - \mu_k\|_2$.
6 **for** $k = 1$ **to** $K$ **do**
7     Select $L$ graphs in $C_k$ closest to $\mu_k$ in moment space: $S_k \subseteq C_k$.
8     Estimate cluster graphon $\hat{W}_k$ using $S_k$ (e.g., via SIGL), and obtain latent positions $\{\eta_i\}$.
9 **return** $\{\hat{W}_k\}_{k=1}^K$ and $\pi$.

---

**Algorithm 2:** Graphon Mixture-Aware Mixup (GMAM)

**Input** : Labeled graphs $\mathcal{D} = \{(G_t, y_t)\}_{t=1}^T$, $y_t \in \{1, \ldots, C\}$; operator $\Phi$ (Alg. 1) run *per class*; mixing coefficient distribution $\lambda \sim \text{Uniform}(0, 0.2)$; target node count $n$; augmentation ratio $r \in (0, 1]$.

**Output** : Augmented set $\widetilde{\mathcal{D}} = \{(G_\lambda^{(m)}, \mathbf{y}_\lambda^{(m)})\}_{m=1}^M$ of size $M = \lceil rT \rceil$.

1 Partition data by class: $\mathcal{D}_i = \{G_t \mid y_t = i\}$ for $i = 1, \ldots, C$.
2 **for** $i = 1$ **to** $C$ **do**
3     $(\{\hat{W}_{i,1}, \ldots, \hat{W}_{i,K_i}\}, \pi_i) \leftarrow \Phi(\mathcal{D}_i)$.
4 $M \leftarrow \lceil rT \rceil$,    $\widetilde{\mathcal{D}} \leftarrow \varnothing$.
5 **for** $m = 1$ **to** $M$ **do**
6     Sample distinct classes $i \neq j$; sample graphs $G_a \in \mathcal{D}_i$, $G_b \in \mathcal{D}_j$.
7     Identify graphons: $\hat{W}_{i,a} \leftarrow \hat{W}_{i,\pi_i(G_a)}$,    $\hat{W}_{j,b} \leftarrow \hat{W}_{j,\pi_j(G_b)}$.
8     Sample $\lambda \sim \text{Uniform}(0, 0.2)$ and set $W_\lambda \leftarrow \lambda \hat{W}_{i,a} + (1 - \lambda)\hat{W}_{j,b}$.
9     Sample latent positions $u_1, \ldots, u_n \overset{\text{i.i.d.}}{\sim} \text{Uniform}[0, 1]$.
10     **for** $1 \le p < q \le n$ **do**
11        Draw $A_\lambda(p, q) \sim \text{Bernoulli}(W_\lambda(u_p, u_q))$ and set $A_\lambda(q, p) \leftarrow A_\lambda(p, q)$.
12     Construct $G_\lambda^{(m)}$ from $A_\lambda$ (and optional node features).
13     Set $\mathbf{y}_\lambda^{(m)} \leftarrow \lambda \, \mathbf{e}_i + (1 - \lambda) \, \mathbf{e}_j$ (one-hot $\mathbf{e}_i$).
14     $\widetilde{\mathcal{D}} \leftarrow \widetilde{\mathcal{D}} \cup \{(G_\lambda^{(m)}, \mathbf{y}_\lambda^{(m)})\}$.
15 **return** $\widetilde{\mathcal{D}}$.

---

---

**Algorithm 3:** Model-Aware Graph Contrastive Learning (MGCL)

> **Input** : Unlabeled graphs $\mathcal{D} = \{G_t\}_{t=1}^T$; shared encoder $\mathcal{E}_\theta$; operator $\Phi$ (Alg. 1); temperature $\tau$; batch size $L$; augmentation rate $r\%$.
> **Output** : Trained encoder parameters $\theta$.

**1** $(\{\hat{W}_1, \ldots, \hat{W}_K\}, \pi) \leftarrow \Phi(\mathcal{D})$ to obtain cluster assignments $C_t \leftarrow \pi(G_t)$.
**2** **for** $epoch = 1, 2, \ldots$ **do**
**3**      Shuffle $\mathcal{D}$ and partition into mini-batches $\mathcal{B}$ of size $L$.
**4**      **for** $batch \ \mathcal{B} = \{G_t\}_{t=1}^L$ **do**
**5**          **foreach** $G_t$ *in* $\mathcal{B}$ **do**
**6**              Let $A_t$ (adjacency) and $X_t$ (features) be inputs.
             `// Graphon-aware augmentation using cluster graphon` $\hat{W}_{C_t}$
**7**              Sample a subset $E_{\text{sel}}$ of $r\%$ node pairs.
**8**              **foreach** $(i, j) \in E_{\text{sel}}$ **do**
**9**                  Draw $\tilde{A}_t(i,j) \sim \text{Bernoulli}(\hat{W}_{C_t}(\eta_i, \eta_j))$; set $\tilde{A}_t(j, i) \leftarrow \tilde{A}_t(i,j)$.
**10**              Set $\tilde{A}_t(i, j) \leftarrow A_t(i, j)$ for all $(i, j) \notin E_{\text{sel}}$.
**11**              Compute embeddings $z_t \leftarrow \mathcal{E}_\theta(A_t, X_t)$ and $\tilde{z}_t \leftarrow \mathcal{E}_\theta(\tilde{A}_t, X_t)$.
         `// Model-aware InfoNCE: negatives only from different clusters`
**12**          Initialize $\mathcal{L}_{\text{batch}} \leftarrow 0$.
**13**          **for** $t = 1$ **to** $L$ **do**
**14**              Define negative index set $\mathcal{N}_t \leftarrow \{t' \in \{1, \ldots, L\} \mid C_{t'} \neq C_t\}$.
**15**              **if** $\mathcal{N}_t = \varnothing$ **then**
**16**                  **continue** (skip or resample batch).
**17**

$$\ell_t \leftarrow -\log \frac{\exp(\text{sim}(z_t, \tilde{z}_t)/\tau)}{\sum\limits_{t' \in \mathcal{N}_t} \exp(\text{sim}(z_t, \tilde{z}_{t'})/\tau)}$$

             $\mathcal{L}_{\text{batch}} \leftarrow \mathcal{L}_{\text{batch}} + \ell_t$.
**18**          $\mathcal{L}_{\text{batch}} \leftarrow \mathcal{L}_{\text{batch}} / |\{t : \mathcal{N}_t \neq \varnothing\}|$.
**19**          Update $\theta$ via gradient step on $\mathcal{L}_{\text{batch}}$.
**20** **return** $\theta$.

---

## A.1. Complexity Analysis of Algorithm 1

The overall time complexity $T_\Phi$ of the algorithm is the sum of its three constituent stages: motif counting, k-means clustering, and per-cluster graphon estimation. The combined complexity is given by:

$$T_\Phi = O\left( \underbrace{T(e_{\max} d_{\max} + n_{\max} d_{\max}^3)}_{\text{Motif Counting}} + \underbrace{I \cdot T \cdot m \log T}_{\text{k-means}} + \underbrace{L \cdot N_e \cdot n_{\max}^2 \log T}_{\text{Graphon Estimation}} \right)$$

Each term in this expression corresponds to a distinct phase of the algorithm. The first term, $O(T(e_{\max} d_{\max} + n_{\max} d_{\max}^3))$, represents the cost of computing the densities for all 9 motifs of size 4 across $T$ graphs, using an efficient graphlet counting algorithm like ORCA (Hočevar & Demšar, 2014). While this term's theoretical worst-case for dense graphs is $O(T \cdot n_{\max}^4)$, making it the apparent bottleneck, our own empirical analysis demonstrates a practical runtime that scales closer to cubic, $O(T \cdot n_{\max}^3)$. This significant practical speed-up is due to an *extremely small leading constant* ($c \approx 2.97 \times 10^{-8}$), a key finding that underpins our method's efficiency (Ramezanpour et al., 2025).

The second term, $O(I \cdot T \cdot m \log T)$, is the standard and well-established complexity for k-means clustering (e.g., via Lloyd's algorithm), which partitions the $T$ motif vectors in $\mathbb{R}^m$ into $K = \log T$ clusters over $I$ iterations. Finally, the third term, $O(L \cdot N_e \cdot n_{\max}^2 \log T)$, accounts for the per-cluster refinement stage. Here, a powerful graphon estimator, such as SIGL (Azizpour et al., 2025), is executed for each of the $K$ clusters on a representative subset of $L$ graphs, requiring $N_e$

training epochs.

In conclusion, while the theoretical complexity suggests motif counting is the most expensive step, its low empirical constant makes the entire pipeline computationally feasible and scalable. This practical efficiency allows our method to effectively operate on many smaller, subsampled graphs, which is a core aspect of its design.

### A.2. Runtime Analysis

In this section, we compute three runtime components for both GMAM and MGCL:

1. The time required to compute motif vectors for all graphs in the dataset (identical for GMAM and MGCL);

2. The time required to perform the first step of our method, which includes clustering and graphon estimation. This step is applied per class in GMAM and once for the entire dataset in MGCL, using the default number of clusters;

3. The time required for actual training and validation, which is independent of the previous steps—that is, motif computation and graphon estimation do not affect training time.

Figure 3 reports these three runtimes for each dataset and setup. As shown, the time required for motif computation, clustering, and graphon estimation is consistently much smaller than the training time, with the gap being especially pronounced in datasets containing a larger number of graphs, such as COLLAB, NCI1, and REDDIT-MULTI-5K.

Combined with the results in Section 4, which show that our method outperforms existing state-of-the-art approaches, the additional cost required for motif-based clustering is well justified.

Finally, we note that in these experiments, we use SIGL for graphon estimation. If runtime were of primary importance, one could instead employ faster estimators such as USVT or SAS.

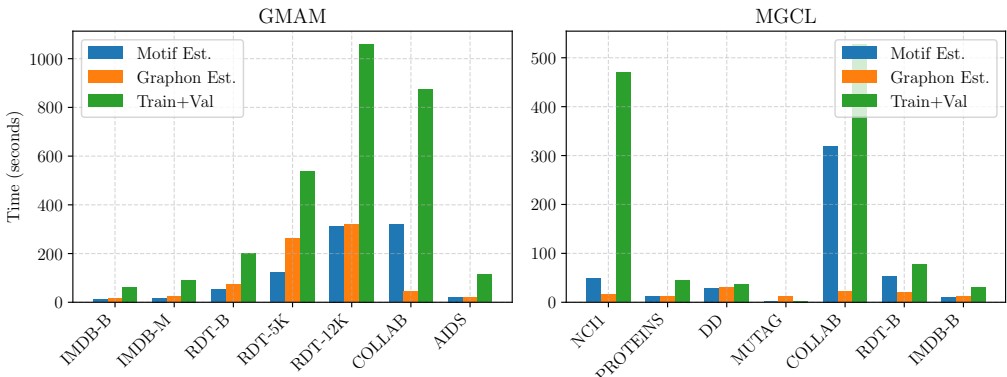

*Figure 3.* Runtime analysis comparing the time required for motif vector computation, graphon estimation, and model training. (a) GMAM (b) MGCL

## B. Proof of Theorem 3.1

This appendix provides the proof for Theorem 3.1 and compares our novel error bound to the classical approach. The proof relies on bounding the difference between the empirical densities using the triangle inequality:

$$|t(F, G_1) - t(F, G_2)| \leq \underbrace{|t(F, W_1) - t(F, W_2)|}_{\text{Graphon Difference}} + \underbrace{|t(F, G_1) - t(F, W_1)|}_{\text{Sampling Error 1}} + \underbrace{|t(F, G_2) - t(F, W_2)|}_{\text{Sampling Error 2}}.$$

The **Graphon Difference** term is bounded by the Counting Lemma (cf. Theorem 10.23 in (Lovász, 2012)):

$$|t(F, W_1) - t(F, W_2)| \leq e(F) d_{\text{cut}}(W_1, W_2) \leq e(F)\epsilon.$$

The **Sampling Error** terms, $|t(F, G_i) - t(F, W_i)|$, are bounded using a novel two-stage analysis that separates the randomness from vertex selection and edge realization. This approach yields a tighter bound than classical single-stage methods. We decompose the total sampling error into two components:

$$|t(F, G) - t(F, W)| \leq \underbrace{|\widehat{t}(F; X) - t(F, W)|}_{\text{Vertex Noise}} + \underbrace{|t(F, G) - \widehat{t}(F; X)|}_{\text{Edge Noise}}$$

where $\widehat{t}(F; X)$ is the conditional expectation of the density given the sampled vertex labels $X = (X_1, \ldots, X_n)$. The following two lemmas bound the vertex and edge noise, respectively.

**Lemma B.1** (Bounding Vertex Noise). *Let $m = \lfloor n/k \rfloor$. For any $\delta_v > 0$,*

$$\Pr\left[|\widehat{t}(F; X) - t(F, W)| \geq \delta_v\right] \leq 2\exp\left(-2m\,\delta_v^2\right).$$

**Lemma B.2** (Bounding Edge Noise). *Conditioned on the vertex labels $X = (X_i)_{i=1}^n$, for any $\delta_e > 0$,*

$$\Pr\left[|t(F, G) - \widehat{t}(F; X)| \geq \delta_e \mid X\right] \leq 2\exp\left(-\frac{n(n-1)\delta_e^2}{2e(F)^2}\right).$$

**Combined Bound** To obtain the total sampling error for one graph, $|t(F, G_i) - t(F, W_i)|$, with a target failure probability $\eta$, we allocate $\eta/2$ to each noise source (vertex and edge). Inverting the bounds in the lemmas above, we set:

$$\delta_v = \sqrt{\frac{1}{2m}\log\frac{4}{\eta}} \quad \text{and} \quad \delta_e = \frac{e(F)}{\sqrt{n(n-1)}}\sqrt{2\log\frac{4}{\eta}}.$$

The total sampling error for one graph is bounded by $\delta_s = \delta_v + \delta_e$ with probability at least $1 - \eta$.

Applying this to both sampling error terms ($|t(F, G_1) - t(F, W_1)|$ and $|t(F, G_2) - t(F, W_2)|$) and using a union bound, we get the final result in Theorem 3.1 with probability at least $1 - 2\eta$:

$$|t(F, G_1) - t(F, G_2)| \leq e(F)\epsilon + 2(\delta_v + \delta_e) = e(F)\epsilon + 2\left(\sqrt{\frac{1}{2m}\log\frac{4}{\eta}} + \frac{e(F)}{\sqrt{n(n-1)}}\sqrt{2\log\frac{4}{\eta}}\right).$$

## B.1. Comparison with the Classical Bound

To highlight the improvement offered by our two-stage approach, we now present the classical single-stage bound and compare them.

### B.1.1. THE CLASSICAL SINGLE-STAGE BOUND

The standard approach (cf. Lemma 4.4 in (Borgs et al., 2008)) treats the empirical motif density $t(F, G)$ as a complex function of $n$ random variables and applies McDiarmid's inequality directly. This results in the following concentration result.

**Lemma B.3** (Classical Concentration of Motif Densities). *Let $F$ be a fixed graph with $k = v(F)$ vertices. Let $G$ be an $n$-vertex graph sampled from a graphon $W$. Then for any $\delta > 0$,*

$$\Pr\left[|t(F, G) - t(F, W)| \geq \delta\right] \leq 2\exp\left(-\frac{n\delta^2}{4k^2}\right).$$

The limitation of this approach is that changing a single vertex can alter many edge probabilities, leading to a loose bound with a factor of $k^2$ in the exponent.

### B.1.2. ASYMPTOTIC COMPARISON

We now compare the sampling error from the classical approach with our novel bound. For a total failure probability $\eta$:

- **Classical Bound:** Inverting the bound in Lemma B.3 yields:

$$\delta_s^{\text{old}}(\eta) = 2k\sqrt{\frac{1}{n}\log\frac{2}{\eta}}$$

- **Novel Bound:** Approximating $m \approx n/k$ and $n(n-1) \approx n^2$, we have:

$$\delta_s^{\text{new}}(\eta) \approx \left(\frac{\sqrt{k}}{\sqrt{2n}} + \frac{e(F)\sqrt{2}}{n}\right)\sqrt{\log\frac{4}{\eta}}$$

The crucial difference lies in their dependence on the motif size, $k$. The classical bound $\delta_s^{\text{old}}$ scales linearly with $k$ ($O(k)$), whereas the dominant term in our novel bound $\delta_s^{\text{new}}$ scales with the square root of $k$ ($O(\sqrt{k})$). As $\sqrt{k}$ grows much more slowly than $k$, our novel bound is substantially tighter for larger motifs.

### B.1.3. NUMERICAL VALIDATION

We validate this theoretical improvement by computing both bounds for all motifs with up to 4 vertices ($k \le 4$), for graph sizes $n$ from 50 to 1000, at a 95% confidence level ($\eta = 0.05$). The results are visualized in Figure 4. The novel bound is uniformly tighter than the classical bound across all tested motifs.

## C. Proof of Theorem 3.2

**Notation.** Let $\mathbf{z}_t = \mathcal{E}(A_t, X_t)$ denote the embedding of graph $t$ obtained by applying the encoder $\mathcal{E}$ to its adjacency matrix $A_t$ and feature matrix $X_t$. Let $\text{sim}(u, v)$ be cosine similarity and define $\theta(u, v) := \frac{\text{sim}(u,v)}{\tau}$. For an anchor graph $t$, write the set of all negatives in standard InfoNCE as $\mathcal{N}_t := \{\, k \in \{1, \ldots, L\} \setminus \{t\} \,\}$, and the *cluster-restricted* negatives in our case as $\widetilde{\mathcal{N}}_t := \{\, k \in \mathcal{N}_t : C_k \ne C_t \,\}$, where $C_i$ is the cluster of graph $i$. For brevity, let $s_k := \theta(\mathbf{z}_t, \mathbf{z}_k')$ and $\tilde{s}_k := \theta(\mathbf{z}_t, \tilde{\mathbf{z}}_k)$ where $\mathbf{z}_t'$ and $\tilde{\mathbf{z}}_t$ are the embedding of the positive sample obtained from a random and graphon-consistent augmentation, respectively. The standard InfoNCE loss for graph $t$ is defined as

$$\ell_{\text{InfoNCE}}(t) = -\log\frac{\exp(\theta(\mathbf{z}_t, \mathbf{z}_t'))}{\sum_{k \in \mathcal{N}_t}\exp(s_k)}. \tag{14}$$

Our proposed cluster-restricted loss is defined as

$$\ell_{\text{cluster}}(t) = -\log\frac{\exp(\theta(\mathbf{z}_t, \tilde{\mathbf{z}}_t))}{\sum_{k \in \widetilde{\mathcal{N}}_t}\exp(\tilde{s}_k)}. \tag{15}$$

Using the natural logarithm and separating the positive-pair term, we can rewrite:

$$\ell_{\text{InfoNCE}}(t) = -\theta(\mathbf{z}_t, \mathbf{z}_t') + \ln\left(\sum_{k \in \mathcal{N}_t} e^{s_k}\right), \tag{16}$$

$$\ell_{\text{cluster}}(t) = -\theta(\mathbf{z}_t, \tilde{\mathbf{z}}_t) + \ln\left(\sum_{k \in \widetilde{\mathcal{N}}_t} e^{\tilde{s}_k}\right). \tag{17}$$

**Proposition C.1** (Lower bound for the cluster-restricted loss)**.** *Let*

$$\ell_{cluster}(t) = -\theta(\mathbf{z}_t, \tilde{\mathbf{z}}_t) + \ln\sum_{k \in \widetilde{\mathcal{N}}_t}\exp\left(\theta(\mathbf{z}_t, \mathbf{z}_k')\right),$$

*where $\widetilde{\mathcal{N}}_t = \{\, k \ne t : C_k \ne C_t \,\}$ and $m_t := |\widetilde{\mathcal{N}}_t|$. Then, for every anchor $t$,*

$$\ell_{cluster}(t) \ge -\theta(\mathbf{z}_t, \tilde{\mathbf{z}}_t) + \frac{1}{m_t}\sum_{k \in \widetilde{\mathcal{N}}_t}\theta(\mathbf{z}_t, \mathbf{z}_k') + \ln m_t. \tag{18}$$

*Proof.* By definition,

$$\ln \sum_{k \in \widetilde{\mathcal{N}}_t} \exp\big(\theta(\mathbf{z}_t, \mathbf{z}'_k)\big) = \ln m_t + \ln\left(\frac{1}{m_t} \sum_{k \in \widetilde{\mathcal{N}}_t} \exp\big(\theta(\mathbf{z}_t, \mathbf{z}'_k)\big)\right).$$

Applying Jensen's inequality to the convex exponential function yields

$$\ln \sum_{k \in \widetilde{\mathcal{N}}_t} \exp\big(\theta(\mathbf{z}_t, \mathbf{z}'_k)\big) \geq \ln m_t + \frac{1}{m_t} \sum_{k \in \widetilde{\mathcal{N}}_t} \theta(\mathbf{z}_t, \mathbf{z}'_k).$$

Substituting into the definition of $\ell_{\text{cluster}}(t)$ gives the claimed lower bound. □

**Proposition C.2** (Lower bound for the standard InfoNCE loss)**.** *Let*

$$\ell_{\text{InfoNCE}}(t) = -\theta(\mathbf{z}_t, \mathbf{z}'_t) + \ln \sum_{k \in \mathcal{N}_t} \exp\big(\theta(\mathbf{z}_t, \mathbf{z}'_k)\big),$$

*where $\mathcal{N}_t = \{\, k \neq t \,\}$ and $n_t := |\mathcal{N}_t| = N - 1$. Define the negative-center score*

$$\bar{\theta}_{-t} \;:=\; \frac{1}{n_t} \sum_{k \in \mathcal{N}_t} \theta(\mathbf{z}_t, \mathbf{z}'_k).$$

*Then, for every anchor t,*

$$\ell_{\text{InfoNCE}}(t) \;\geq\; -\theta(\mathbf{z}_t, \mathbf{z}'_t) \;+\; \bar{\theta}_{-t} \;+\; \ln n_t. \tag{19}$$

*Proof.* Apply Jensen's inequality to the convex exponential:

$$\ln \sum_{k \in \mathcal{N}_t} e^{\theta(\mathbf{z}_t, \mathbf{z}'_k)} = \ln n_t + \ln\left(\frac{1}{n_t} \sum_{k \in \mathcal{N}_t} e^{\theta(\mathbf{z}_t, \mathbf{z}'_k)}\right) \geq \ln n_t + \frac{1}{n_t} \sum_{k \in \mathcal{N}_t} \theta(\mathbf{z}_t, \mathbf{z}'_k) = \ln n_t + \bar{\theta}_{-t}.$$

Substitute into the definition of $\ell_{\text{InfoNCE}}(t)$ to obtain (19). □

# D. Graphon estimation

Here we explain the details of SIGL. The goal is to estimate an unknown graphon $\omega : [0,1]^2 \to [0,1]$, given a set of graphs $\mathcal{D} = \{\mathbf{A}_t\}_{t=1}^M$ sampled from it. Since using the Gromov-Wasserstein (GW) (Mémoli, 2011) distance is computationally infeasible for large graphs, the SIGL framework (Azizpour et al., 2025) proposes a scalable three-step procedure:

**Step 1: Sorting nodes via latent variable estimation**   To align all graphs to a common node ordering (which is crucial for consistent estimation), SIGL estimates latent variables $\hat{\boldsymbol{\eta}}_t = \{\hat{\eta}_i\}_{i=1}^{n_t}$ for each graph $G_t$ using a Graph Neural Network (GNN):

$$\hat{\boldsymbol{\eta}}_t = g_{\phi_1}(\mathbf{A}_t, \mathbf{Y}_t), \quad \text{where } \mathbf{Y}_t \sim \mathcal{N}(0, 1)$$

An auxiliary graphon $h_{\phi_2}$ modeled by an Implicit Neural Representation (INR) maps pairs of latent variables to edge probabilities:

$$h_{\phi_2}(\hat{\boldsymbol{\eta}}_t(i), \hat{\boldsymbol{\eta}}_t(j)) \approx \mathbf{A}_t(i, j)$$

The latent variables and auxiliary graphon are jointly trained by minimizing the mean squared error to get $\phi = \{\phi_1 \cup \phi_2\}$:

$$\mathcal{L}(\phi) = \sum_{t=1}^M \frac{1}{n_t^2} \sum_{i,j=1}^{n_t} \big[\mathbf{A}_t(i,j) - h_{\phi_2}(\hat{\boldsymbol{\eta}}_t(i), \hat{\boldsymbol{\eta}}_t(j))\big]^2$$

A sorting permutation $\hat{\pi}$ is defined based on the learned latent variables:

$$\hat{\boldsymbol{\eta}}_t(\hat{\pi}(1)) \geq \hat{\boldsymbol{\eta}}_t(\hat{\pi}(2)) \geq \cdots \geq \hat{\boldsymbol{\eta}}_t(\hat{\pi}(n_t))$$

In a nutshell, this permutation sorts the latent variables from 0 to 1. The graphs are reordered accordingly to produce sorted adjacency matrices $\hat{\mathbf{A}}_t$.

**Step 2: Histogram Approximation**   For each sorted graph $\hat{\mathbf{A}}_t$, a histogram $\hat{\mathbf{H}}_t \in \mathbb{R}^{k \times k}$ is computed using average pooling with window size $h$:

$$\hat{\mathbf{H}}_t(i, j) = \frac{1}{h^2} \sum_{s_1=1}^{h} \sum_{s_2=1}^{h} \hat{\mathbf{A}}_t \left( (i-1)h + s_1, (j-1)h + s_2 \right)$$

This results in a new dataset $\mathcal{I} = \{\hat{\mathbf{H}}_t\}_{t=1}^{M}$, providing discrete, noisy views of the unknown graphon $\omega$.

**Step 3: Training the Graphon INR**   The final step constructs a supervised dataset $\mathcal{C}$ from all histograms, where each point corresponds to a coordinate-value triple:

$$\mathcal{C} = \left\{ \left( \frac{i}{k_t}, \frac{j}{k_t}, \hat{\mathbf{H}}_t(i, j) \right) : i, j \in \{1, \ldots, k_t\}, \, t \in \{1, \ldots, M\} \right\}$$

A second INR structure $f_\theta : [0, 1]^2 \to [0, 1]$ is then trained to regress the graphon values by minimizing the MSE:

$$\mathcal{L}(\theta) = \sum_{(x,y,z) \in \mathcal{C}} (f_\theta(x, y) - z)^2$$

This scalable approach enables two things: 1) the estimation of a continuous graphon $\omega$ using large-scale graph data without relying on costly combinatorial metrics like the GW distance, and 2) the estimation of the latent variables given an input graphs, i.e., an inverse mapping $\mathcal{W}^{-1} : \mathbf{A} \to \boldsymbol{\eta}$.

# E. Experimental details.

### E.1. Ground truth graphons

In Table 4, we provide the mathematical definition of the graphon used in Section 4 for the synthetic experiments.

*Table 4.* Ground truth graphons.

|   | $\omega(x, y)$ |
|---|---|
| 0 | $xy$ |
| 1 | $\exp(-(x^{0.7} + y^{0.7}))$ |
| 2 | $\frac{1}{4}(x^2 + y^2 + \sqrt{x} + \sqrt{y})$ |
| 3 | $\frac{1}{2}(x + y)$ |
| 4 | $(1 + \exp(-2(x^2 + y^2)))^{-1}$ |
| 5 | $(1 + \exp(-\max\{x, y\}^2 - \min\{x, y\}^4))^{-1}$ |
| 6 | $\exp(-\max\{x, y\}^{0.75})$ |

**GW distance of ground truth graphons.**   In Table 5, we report pairwise Gromov–Wasserstein (GW) distances between ground-truth graphons, serving as a practical surrogate for the cut distance via its relaxation to GW distance on step functions (Xu et al., 2021).

|   | 1 | 2 | 3 | 4 | 5 | 6 |
|---|---|---|---|---|---|---|
| **0** | 0.133 | 0.265 | 0.264 | 0.530 | 0.418 | 0.271 |
| **1** |  | 0.180 | 0.189 | 0.433 | 0.308 | 0.173 |
| **2** |  |  | 0.024 | 0.270 | 0.175 | 0.111 |
| **3** |  |  |  | 0.275 | 0.190 | 0.126 |
| **4** |  |  |  |  | 0.139 | 0.284 |
| **5** |  |  |  |  |  | 0.162 |

*Table 5.* Pairwise GW distances between groundtruth graphons.

### E.2. Hyper-parameters

**Graphon estimation**  The hyperparameters used to estimate the graphon with SIGL across its three steps, as described in the previous section, are as follows for both mixup and graph level tasks. We use the `Adam` optimizer (Kingma, 2015) with a learning rate of $lr = 0.01$ for both Step 1 and Step 3, running for 40 and 20 epochs, respectively. In Step 1, the batch size is set to 1 graph, while in Step 3, each batch includes 1024 data points from $\mathcal{C}$. In Step 1, the GNN, $g_{\phi_1}$ comprises two consecutive graph convolutional layers, each followed by a ReLU activation function. All convolutional layers use 8 hidden channels. The INR structures in Step 1 ($h_{\phi_2}$) and Step 3 ($f_\theta$) each have 3 layers with 20 hidden units per layer and use a default frequency of 10 for the $\sin(.)$ activation function.

**Mixup**  To ensure a fair comparison, we use the same hyperparameters for model training and the same architecture across vanilla models and other baselines. Also, we conduct the experiments using the same hyperparameter values as in Han et al. (2022). For graph classification tasks, we employ the `Adam` optimizer with an initial learning rate of 0.01, which is halved every 100 epochs over a total of 800 epochs. The batch size is set to 128. The dataset is split into training, validation, and test sets in a 7:1:2 ratio. The best test epoch is selected based on validation performance, and test accuracy is reported over eight runs with the same `seed` used in Han et al. (2022).

We generate 20% more graphs for training. The graphons are estimated from the training graphs, and we use different $\lambda$ values in the range [0.1, 0.2] to control the strength of mixing in the generated synthetic graphs. For graphon estimation using SIGL and IGNR, we use 20% of the training data per class to estimate the graphon. The new graphs are generated with the average number of nodes as defined for the primary G-Mixup case, which is identified as the optimal size. All methods are evaluated using the same random seeds.

**MGCL**  To train the encoder $\mathcal{E}_\theta$, we follow the configuration from GraphCL (You et al., 2020). We use the `Adam` optimizer with a learning rate of $lr = 0.001$, training the encoder for 20 epochs. The encoder is implemented as a 3-layer GIN (Xu et al., 2019) network, with each layer consisting of 32 hidden units followed by a ReLU activation. A final linear projection head maps the output to a 32-dimensional graph-level representation. For a fair comparison, we set the resampling ratio to $r = 20\%$, consistent with other methods. All methods are evaluated using the same random seeds.

### E.3. Dataset Details

In Table 6, we report the statistics of all datasets from TUDataset (Morris et al., 2020) used in our real-world experiments.

*Table 6.* Benchmark datasets statistics.

| Statistic | Biochemical Molecules | | | | | Social Networks | | | |
|---|---|---|---|---|---|---|---|---|---|
| | NCI1 | PROTEINS | DD | MUTAG | AIDS | COLLAB | RDT-B | RDT-M5K | IMDB-B |
| #Graphs | 4,110 | 1,113 | 1,178 | 188 | 2,000 | 5,000 | 2,000 | 4,999 | 1,000 |
| Avg. #Nodes | 29.87 | 39.06 | 284.32 | 17.93 | 15.69 | 74.5 | 429.6 | 508.8 | 19.8 |
| Avg. #Edges | 32.30 | 72.82 | 715.66 | 19.79 | 16.20 | 2457.78 | 497.75 | 594.87 | 96.53 |
| #Classes | 2 | 2 | 2 | 2 | 2 | 3 | 2 | 5 | 2 |

Note that several graph-level datasets lack explicit node attributes. In such cases, one-hot encoding of node degrees is commonly used to construct node features.

### E.4. Baseline models.

Here we have a small description of the baselines used in the synthetic, mixup, and GCL experiments.

#### E.4.1. GRAPH EMBEDDING BASELINE DETAILS

This section provides further details on the baseline graph embedding methods used for comparison in the synthetic clustering experiment (Section 4.1). We compare our moment vectors against five widely-used graph representation methods. For methods that produce node-level embeddings (GCN, GIN, DeepWalk, Spectral), we obtain a graph-level embedding by applying a global mean-pooling operation over all node embeddings in the graph. The target embedding dimension for all baselines was set to 32.

- Moment Embedding (Ours): This is our proposed method, where each graph is represented by a 9-dimensional vector of its empirical motif densities for all connected motifs up to 4 nodes.

- Graph Convolutional Network (GCN): A popular Graph Neural Network (GNN) architecture that learns node representations by iteratively aggregating feature information from local neighborhoods (Kipf & Welling, 2017). As our clustering task is unsupervised, the GCN is not trained; we use a randomly initialized network to generate embeddings. This standard approach tests the intrinsic structural representation power of the architecture itself.

- Graph Isomorphism Network (GIN): A powerful GNN model proven to be as discriminative as the Weisfeiler-Leman test for graph isomorphism (Xu et al., 2019). Similar to our use of GCN, the GIN model's weights are kept random, allowing it to serve as an unsupervised feature extractor without training on downstream labels.

- Graph2Vec: An unsupervised whole-graph embedding method inspired by natural language processing. It treats graphs as documents and rooted subgraphs as "words," learning representations that capture structural similarities between graphs (Narayanan et al., 2017).

- DeepWalk: A pioneering node embedding technique that learns latent representations by applying language modeling techniques (Skip-Gram) to sequences of nodes generated from truncated random walks on the graph (Perozzi et al., 2014).

- Spectral Embedding: A classical approach that utilizes the eigenvectors of the graph's Laplacian matrix. The first few non-trivial eigenvectors form a low-dimensional representation of the nodes, capturing the graph's global connectivity structure (Ng et al., 2001).

- DisenGCN: A disentangled graph convolutional network that decomposes node representations into multiple latent factors by routing neighborhood information into factor-specific channels, aiming to capture different generative mechanisms underlying graph connectivity (Ma et al., 2019).

### E.4.2. MIXUP BASELINES.

- *Vanilla*, a baseline with no augmentation;

- *DropEdge* (Rong et al., 2020), which uniformly removes a fraction of edges;

- *DropNode* (Feng et al., 2020), which randomly drops a portion of nodes;

- *Subgraph* (You et al., 2020), which extracts random-walk-based subgraphs;

- *M-Mixup* (Wang et al., 2021), which linearly interpolates graph-level representations;

- *SubMix* (Yoo et al., 2022), which mixes random subgraphs of graph pairs;

- *G-Mixup* (Han et al., 2022), which performs class-level graph Mixup by interpolating graphons from different classes;

- *S-Mixup* (Ling et al., 2023), which employs soft alignment for graph interpolation;

- *SIGL* (Azizpour et al., 2025), which replaces the original graphon estimator in G-Mixup with SIGL; and

- *MomentMixup* (Ramezanpour et al., 2025), which mixes motif moment vectors and generates new samples from the mixed vector.

### E.4.3. GCL BASELINES.

- InfoGraph (Sun et al., 2020): A variant of DGI (Veličković et al., 2019) that maximizes mutual information between graph-level and substructure representations at multiple scales.

- GraphCL (You et al., 2020): Learns representations via predefined augmentations such as edge perturbation, node dropping, and attribute masking.

- MVGRL (Hassani & Khasahmadi, 2020): Performs contrastive learning between structural views, e.g., adjacency and diffusion representations.

*Table 7.* Results of the ablation study for the first 15 motifs. Clustering accuracy (%) is shown for K-Means and Theory-Based methods across two dataset settings. The performance largely saturates after $k = 9$.

| # Motifs | Varying Size Accuracy (%) | | Fixed Size Accuracy (%) | |
| --- | --- | --- | --- | --- |
| $(k)$ | K-Means | Theory-Based | K-Means | Theory-Based |
| 1 | 70.0 | 74.3 | 74.7 | 74.7 |
| 2 | 67.1 | 77.1 | 74.7 | 76.9 |
| 3 | 78.6 | 80.0 | 79.1 | 82.3 |
| 4 | 78.6 | 80.0 | 79.1 | 82.7 |
| 5 | 78.6 | 80.0 | 79.1 | 82.9 |
| 6 | 78.6 | 80.0 | 79.1 | 83.0 |
| 7 | 78.6 | 80.0 | 79.1 | 83.0 |
| 8 | 78.6 | 80.0 | 79.1 | 83.0 |
| 9 | 80.0 | 81.4 | 79.4 | 83.6 |
| 10 | 80.0 | 81.4 | 79.4 | 83.6 |
| 11 | 80.0 | 81.4 | 79.4 | 83.6 |
| 12 | 80.0 | 81.4 | 79.4 | 83.6 |
| 13 | 80.0 | 81.4 | 79.4 | 83.6 |
| 14 | 80.0 | 81.4 | 79.4 | 83.6 |
| 15 | 80.0 | 81.4 | 79.4 | 83.6 |

- JOAO (You et al., 2021): Uses min-max optimization to adaptively select augmentations during training.

- Ad-GCL (Suresh et al., 2021): An adversarial training–based GCL framework that introduces an edge-perturbation process designed as an attack to improve robustness.

- AutoGCL (Yin et al., 2022): Employs learnable view generators with auto-augmentation for adaptive, label-preserving samples.

- SimGRACE (Xia et al., 2022): Replaces graph augmentations with encoder-level noise to enforce consistency.

- RGCL (Li et al., 2022): Generates rationale-aware views by identifying substructures most relevant for discrimination.

- DRGCL (Ji et al., 2024): Learns dimensional rationales and applies bi-level meta-learning to mitigate confounding noise.

### E.5. Compute resources

All experiments were conducted on a server running Ubuntu 20.04.6 LTS, equipped with an AMD EPYC 7742 64-Core Processor and an NVIDIA A100-SXM4-80GB GPU with 80GB of memory. For model development, we utilized PyTorch version 1.13.1, along with PyTorch Geometric version 2.3.1, which also served as the source for all datasets used in our study.

## F. Additional Experiments.

### F.1. Ablation Study on Number of Motifs

To justify our choice of using motifs up to 4 nodes (a 9-dimensional feature vector) in the main experiment, we conduct an ablation study to analyze the effect of the number of motifs on clustering performance. We extend the experimental setup from Section 4.1 by computing densities for all 30 connected motifs with up to 5 nodes. We then iteratively evaluate the clustering accuracy of our methods by using an increasing number of motifs.

The results for the first 15 motifs are presented in Table 7. A clear trend emerges: accuracy rises sharply with the inclusion of the first few motifs, but then exhibits a long plateau of nearly constant performance. This indicates that some motifs are significantly more important for classification than others. A particularly notable increase occurs between $k = 8$ and $k = 9$ motifs in the *Varying* size setting, where accuracy jumps from 80.0% to 81.4% and then flatlines. This specific jump highlights the discriminative power of the 9th motif (the 4-cycle) and suggests that this initial set of motifs captures the most critical structural differences between the graphon families. This study confirms that the initial 9 motifs provide the vast majority of the useful information. The minimal performance gain from adding more motifs (from $k = 10$ to $k = 15$ and

*Table 8.* Graphon estimation accuracy measured by Gromov–Wasserstein (GW) loss for the mixture setting compared to the oracle single-graphon baseline, across varying-size and fixed-size synthetic datasets.

| Cluster ID | Varying Size GW Loss | | Fixed Size GW Loss | |
| --- | --- | --- | --- | --- |
| | Ours | Baseline | Ours | Baseline |
| 0 | 0.0433 | 0.0152 | 0.0340 | 0.0152 |
| 1 | 0.0381 | 0.0258 | 0.0198 | 0.0212 |
| 2 | 0.0272 | 0.0258 | 0.0294 | 0.0205 |
| 3 | 0.0238 | 0.0165 | 0.0293 | 0.0246 |
| 4 | 0.0254 | 0.0212 | 0.0343 | 0.0168 |
| 5 | 0.0280 | 0.0205 | 0.0301 | 0.0258 |
| 6 | 0.0278 | 0.0246 | 0.0116 | 0.0165 |

beyond) does not justify the increased computational cost. This validates our use of the 9-dimensional moment vector in the main paper as an effective and efficient choice.

We note two regimes in which a truncated motif family can fail to separate components of the mixture. First, when distinct graphons agree on all low-order motif densities but differ in higher-order structure, the truncated family becomes uninformative and the family must be extended to motifs of size $\geq 5$. Second, in the small-graph regime, sampling noise in the motif estimates (Theorem 3.1) can dominate the inter-graphon signal and obscure separation even when the family is in principle sufficient (see Appendix F.2 and Appendix G.4).

### F.2. Effect of Graph Size on Moment Concentration

Theorem 3.1 indicates that the sampling-error term in the empirical motif density decreases with graph size $n$, so smaller graphs should produce noisier moment vectors and weaker cluster separation. While Figure 2 illustrates this effect in aggregate, the size of each individual graph is not shown per data point. To examine the role of graph size directly, we conduct a controlled experiment in which graph size is the only varying factor. We sample graphs from 3 distinct graphons and, for each graphon, draw 40 graphs at each of the node sizes $n \in \{5, 10, 100, 300\}$. For every sampled graph, we compute its empirical moment vector over the same motif family used in the main experiments, and we visualize a low-dimensional projection of these vectors grouped by graph size. The results are shown in Figure 5. For very small graphs ($n = 5, 10$), the moment vectors exhibit a large spread and the three graphon groups overlap substantially, consistent with the high sampling variance predicted by Theorem 3.1. As the graph size increases ($n = 100, 300$), the moment vectors concentrate tightly around their graphon-specific centers and the three groups become clearly separated. This directly confirms that the separability of motif-based embeddings improves with graph size, and it makes explicit the source of the more diffuse clusters observed in the varying-size setting of Figure 2(a).

### F.3. Graphon Estimation Quality

Since the ground-truth graphons are unknown for real-world datasets, we cannot directly evaluate graphon estimation accuracy in those settings. To address this, we added a synthetic experiment where the true graphons are available. In this experiment, we compare the estimation quality in two scenarios: (i) the standard setting with a single underlying graphon estimated from observed graphs generated from that single graphon, and (ii) our mixture setting, where graphs are generated from multiple underlying graphons, and we estimate a separate graphon for each cluster using our Algorithm 1.

As illustrated in Table 8, our method consistently recovers the underlying graphons with low loss across different dataset settings. For instance, in the varying-size dataset, we achieve a Gromov-Wasserstein (GW) loss as low as 0.0238 (Cluster 3).

When compared to the oracle single-graphon baseline, our mixture model's performance is highly competitive. For example, our estimation for Cluster 2 in the varying-size setting yields a GW loss of 0.0272, which is close to the corresponding single-graphon baseline of 0.0258. This confirms that our cluster-wise estimation strategy effectively disentangles the mixture without significantly compromising estimation quality.

### F.4. Ablation Study on the number of Clusters

#### F.4.1. SYNTHETIC SETUP

To assess the quality of the clusters found by our approach and to study the effect of tuning the number of mixture components, $k$, we conducted an ablation study on our synthetic dataset, which has 7 ground-truth underlying mixtures. We

*Table 9.* Adjusted Rand Index (ARI) for different numbers of clusters $k$ across synthetic datasets. Bold values indicate the best result in each column.

| $k$ | Varying Size | | Fixed Size | |
|---|---|---|---|---|
| | Moment Emb. | GCN Emb. | Moment Emb. | GCN Emb. |
| 2 | 0.2048 | 0.2555 | 0.2048 | 0.2578 |
| 3 | 0.3616 | 0.3860 | 0.3616 | 0.4316 |
| 4 | 0.5463 | 0.4498 | 0.5463 | 0.4993 |
| 5 | 0.6296 | **0.4764** | 0.6223 | **0.5709** |
| 6 | **0.7166** | 0.4242 | **0.6889** | 0.5133 |
| 7 | 0.6709 | 0.3981 | 0.6444 | 0.4842 |
| 8 | 0.6650 | 0.4215 | 0.6756 | 0.4599 |
| 9 | 0.6276 | 0.4189 | 0.6463 | 0.4256 |
| 10 | 0.6953 | 0.4158 | 0.6073 | 0.4076 |

measured the cluster quality using the **Adjusted Rand Index (ARI)**, where a higher score indicates a better match to the ground truth partitioning.

Table 9 shows this comparison for our Moment Embeddings and the GCN baseline across various values of $k$ (from 2 to 10). We omitted other baselines for space, as GCN was the strongest performing baseline, with others having significantly lower ARI scores.

The results show that our Moment Embedding's ARI score **peaks at** $k = 6$ for both the varying-size and fixed-size graph datasets. This result is consistent with the ground truth of 7 mixtures. The peak at $k = 6$ rather than $k = 7$ is expected because the ground-truth Graphons 2 and 3 are structurally very similar (Gromov-Wasserstein distance of $0.024$, as noted in Section 4.1), and our method successfully groups these similar structures.

In contrast, the GCN baseline's cluster quality peaks at $k = 5$, and its peak ARI (e.g., $0.5709$ for fixed size) is significantly lower than our method's peak (e.g., $0.7166$ for varying size). This demonstrates that the GCN method fails to accurately recover the underlying graph structure.

### F.4.2. GMAM

We conduct a similar ablation in the GMAM setting by varying the number of clusters used to estimate the underlying models for each graph class. As shown in Figure 6, using more than one cluster consistently improves performance on both IMDB-B and IMDB-MULTI.

For IMDB-B, selecting between 2 and 9 clusters consistently outperforms the single-cluster setting, with performance peaking at the default value of 6 clusters. In IMDB-MULTI, using 2–10 clusters always matches or exceeds the accuracy obtained with a single cluster, and the best performance occurs at 4 clusters—slightly lower than the default of 6—suggesting that 4 clusters adequately capture the underlying generative models for this dataset.

In both datasets, using 10 clusters results in a slight decrease in accuracy, indicating that this number may be larger than necessary and could lead to over-fragmentation of the data, thereby overlooking shared structural patterns among graphs.

### F.4.3. MGCL

As mentioned in the Methods section, in order to match the size of the data, we use $K = \log T$ for the operator $\Phi$. Here we evaluate this number on three datasets with within the MGCL framework. In this section, we vary the number of clusters to evaluate its impact on overall performance.

We vary the number of clusters from 1 to 10. For each value, we repeat the MGCL experiment across the same 10 trials using identical data partitions and report the average accuracy.

The results are presented in Figure 7 for the MUTAG, IMDB-BINARY, and PROTEINS datasets. A key observation is that using a single cluster-i.e., estimating one graphon for the entire dataset-leads to a drop in performance across all three datasets. In the IMDB-BINARY dataset, using 8 clusters yields better performance than the default setting of 7 clusters.

However, using between 6 and 8 clusters consistently results in an average accuracy above 72.0%, suggesting that this range reasonably approximates the number of underlying models. A similar trend is observed in the PROTEINS dataset, where 9 clusters yield better performance than 8. For the MUTAG dataset, the highest average accuracy is achieved with the default setting of 6 clusters. These findings highlight that estimating multiple models improves performance. Still, the number of clusters remains a hyperparameter that should be tuned based on the dataset's characteristics, such as its variability and heterogeneity.

### F.5. Ablation on the Number of Mixture Components

To thoroughly analyze the robustness of our mixture estimation, we conducted an ablation study on the number of underlying mixture components.

We constructed synthetic datasets with an increasing number of underlying mixtures, ranging from $k = 2$ to $k = 7$. We evaluated the performance using both the Adjusted Rand Index (ARI) and clustering accuracy. To ensure clarity, we present the ablation results in two separate tables: one for *varying-size graphs* (Table 10) and one for *fixed-size graphs* (Table 11).

*Table 10.* Ablation on Number of Mixtures (Varying Size).

| Num. Mixtures | ARI | | Accuracy | |
|---|---|---|---|---|
| | Moment Emb. | GCN | Moment Emb. | GCN |
| 2 | 1.0000 | 0.6383 | 1.0000 | 0.9000 |
| 3 | 1.0000 | 0.6633 | 1.0000 | 0.8667 |
| 4 | 0.6725 | 0.4707 | 0.7750 | 0.6500 |
| 5 | 0.7539 | 0.5977 | 0.8200 | 0.7200 |
| 6 | 0.8027 | 0.5517 | 0.8500 | 0.6833 |
| 7 | 0.7166 | 0.4375 | 0.8000 | 0.5857 |

*Table 11.* Ablation on Number of Mixtures (Fixed Size).

| Num. Mixtures | ARI | | Accuracy | |
|---|---|---|---|---|
| | Moment Emb. | GCN | Moment Emb. | GCN |
| 2 | 0.9406 | 0.7911 | 0.9850 | 0.9450 |
| 3 | 0.9703 | 0.8208 | 0.9900 | 0.9367 |
| 4 | 0.6445 | 0.5021 | 0.7450 | 0.6875 |
| 5 | 0.7332 | 0.6298 | 0.7960 | 0.7520 |
| 6 | 0.7865 | 0.5283 | 0.8300 | 0.6833 |
| 7 | 0.6899 | 0.5222 | 0.7857 | 0.6471 |

These results demonstrate that our moment-based embeddings consistently outperform the GCN baseline across all levels of mixture complexity. In simpler cases ($k = 2$ and $k = 3$ mixtures), our method achieves perfect or near-perfect clustering. Even when scaled to the maximum tested complexity ($k = 7$ mixtures), our method maintains strong performance, whereas the baseline method's performance degrades substantially.

### F.6. False negative reduction.

To evaluate the effect of clustering on the rate of false negatives, we define the True Negative to False Negative Ratio (TFR).

To compute this metric, in each data batch, we examine the negative samples relative to a graph $i$. Among these, the negative samples that share the same class as graph $i$ are considered false negatives, while those with a different class are treated as true negatives. Mathematically, TFR is defined as TFR $= \frac{1}{B} \sum_{i=1}^{B} \frac{|\mathcal{TN}(i)|}{\max\{1, |\mathcal{FN}(i)|\}}$. Note that in InfoNCE-based methods, all graphs in the batch except graph $i$ itself are treated as negative samples. However, in MGCL, the set of negative samples is smaller, as we exclude graphs from the same cluster as $i$. Although this reduced set naturally results in fewer false negatives, computing the relative ratio of true negatives to false negatives (TFR) ensures a fair comparison across methods. We compute the TFR for each graph in the batch and then average it across all graphs in the dataset. As shown in Figure 8,

this metric increases across all datasets compared to the baseline (which represents all InfoNCE-based methods). This also provides evidence that motif-based clustering helps uncover the true underlying structure of the data.

### F.7. Contributions of GIA and CAL in MGCL

MGCL combines two distinct mechanisms: graphon-informed augmentation (GIA), which generates positive samples from the anchor's inferred graphon, and the cluster-aware loss (CAL), which restricts negatives to graphs from different clusters. To isolate their individual contributions, we perform an ablation that adds these components one at a time on top of the standard GraphCL baseline:

- **GraphCL**: random augmentation with the standard InfoNCE loss.

- **MGCL w/o CAL**: GraphCL augmented with GIA only (graphon-informed positives, standard InfoNCE negatives).

- **MGCL**: GraphCL with both GIA and CAL.

Table 12 reports the results on three datasets. Adding GIA alone consistently improves over GraphCL, and adding CAL on top of GIA yields a further consistent gain on all three datasets. This decomposition shows that the two mechanisms contribute complementary improvements, and it aligns with the theoretical decomposition of the lower bound in Theorem 3.2 (Theorem 3.2) as described in Appendix C: GIA strengthens the alignment term $\mathrm{sim}(z_t, \tilde{z}_t)$ by producing more semantically faithful positive pairs, while CAL affects the remaining two terms by restricting negatives to different clusters and pushing the negative centroid farther from the anchor.

*Table 12.* Component ablation for MGCL. Starting from GraphCL, adding graphon-informed augmentation (GIA) and then the cluster-aware loss (CAL) each yields consistent gains, isolating their individual contributions.

| Method | PROTEINS | MUTAG | IMDB-B |
|---|---|---|---|
| GraphCL (random aug. + InfoNCE) | $74.39 \pm 0.4$ | $86.80 \pm 1.3$ | $71.14 \pm 0.4$ |
| MGCL w/o CAL (GraphCL + GIA) | $74.79 \pm 0.6$ | $88.29 \pm 1.7$ | $71.38 \pm 0.8$ |
| MGCL (GraphCL + GIA + CAL) | $75.54 \pm 0.3$ | $90.03 \pm 1.3$ | $72.28 \pm 0.5$ |

### F.8. Underlying models

In Figure 9, we present the estimated graphons for the COLLAB and IMDB-BINARY datasets, each corresponding to a cluster identified by our framework in an unsupervised manner. For instance, the estimated graphons display diverse structural patterns for COLLAB: Cluster 8 exhibits a block-like structure similar to a two-community stochastic block model (SBM) with an imbalanced size ratio, Clusters 2 and 6 display nearly uniform connectivity, suggesting dense or complete graph structures, and Cluster 10 reveals a heavy-tailed pattern indicative of power-law behavior. Although some similarities exist among certain models, the overall variability emphasizes the presence of multiple distinct generative mechanisms within the dataset. This heterogeneity demonstrates the limitations of using a single fixed or random augmentation strategy, as commonly adopted in existing GCL and Mixup methods.

Furthermore, when estimating models within each class for mixup applications, we observe diverse graphons both within and across classes, as illustrated in Figure 10.

## G. Scope and Limitations.

### G.1. Small / high-variance graphs.

As made explicit by Theorem 3.1, the finite-sample error in empirical motif densities is larger for smaller graphs, so the small-graph regime is inherently harder for any moment-based method. Our pipeline mitigates this by clustering in moment space and estimating each component from groups of similar graphs, which yields a cluster-level averaging effect rather than relying on any individual graph. Robustness, therefore, depends more strongly on the number of available graphs and the stability of the estimated motif statistics than on the size of any single graph. We characterized this behavior empirically in Appendix F.2.

### G.2. Dense-graphon regime.

Our framework is developed within the classical dense-graphon setting, in which graphons arise as limit objects of dense graph sequences. Accordingly, the theoretical results of Section 3.1, in particular Theorem 3.1, and the graphon-mixture interpretation should be understood within this regime, and graphon mixtures should not be interpreted as a universally valid generative model for all graph data without qualification. In particular, the present theory is not intended to fully cover highly sparse, large-scale networks.

### G.3. Relaxation to sparse models.

The dense-graphon assumption can in principle be relaxed. For sparse random graph models of the form $\rho_n W$, one can work with suitably normalized motif densities, and the decomposition underlying Theorem 3.1 continues to hold with sparsity-dependent concentration terms. This is consistent with the literature on sparse graph limits and sparse exchangeable graphs (Borgs et al., 2019; 2018b;a), where $L^p$ graphons and graphex-type models provide natural extensions beyond the dense setting. We regard the dense formulation adopted here as a clean and theoretically tractable first setting; a sparse latent model with a corresponding sparse graphon estimator would provide a closer modeling match for sparse datasets and is a natural direction for future work.

### G.4. Sparse benchmarks.

Several of the benchmark datasets used in our experiments (e.g., NCI1 and MUTAG) contain relatively sparse graphs. In these cases, the estimated dense graphon is best viewed as a cluster-level approximation that matches the relevant motif (moment) structure of each subpopulation, rather than as an exact data-generating model. Empirically, our method remains effective on these datasets, which indicates that the estimated dense graphons still capture useful cluster-level structure even when the dense-graph assumption is not strictly satisfied. We emphasize that the central mechanism of our framework, motif-based identification of heterogeneous graph populations and the use of that structure in downstream learning, does not depend on the dense-graph asymptotics holding exactly for every finite dataset.

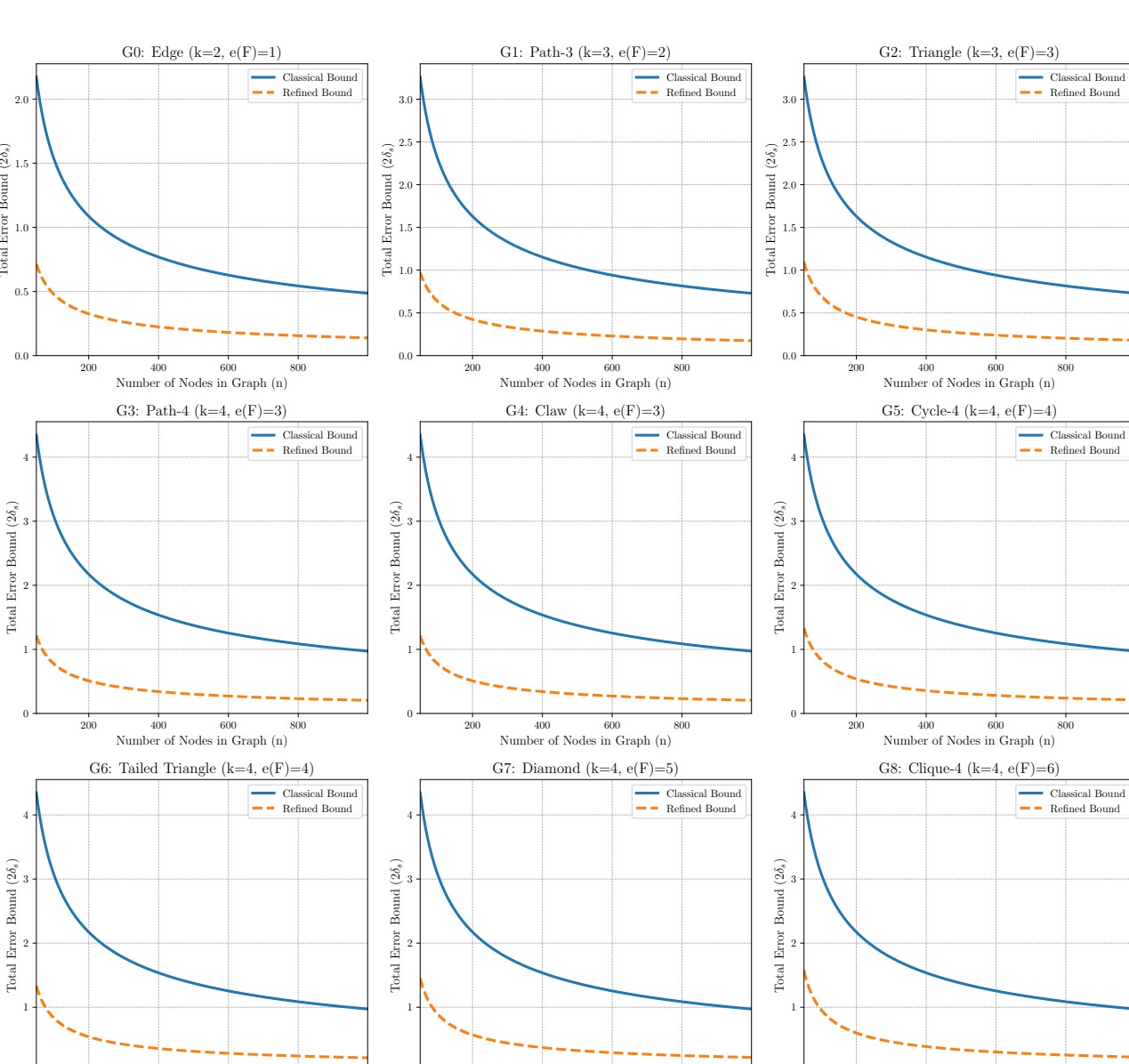

*Figure 4.* Comparison of the total error bounds ($2\delta_s$) for the classical (solid blue) and novel (dashed orange) approaches. The novel bound is consistently tighter, with the gap widening for motifs with more vertices ($k$), confirming its superior $O(\sqrt{k})$ scaling.

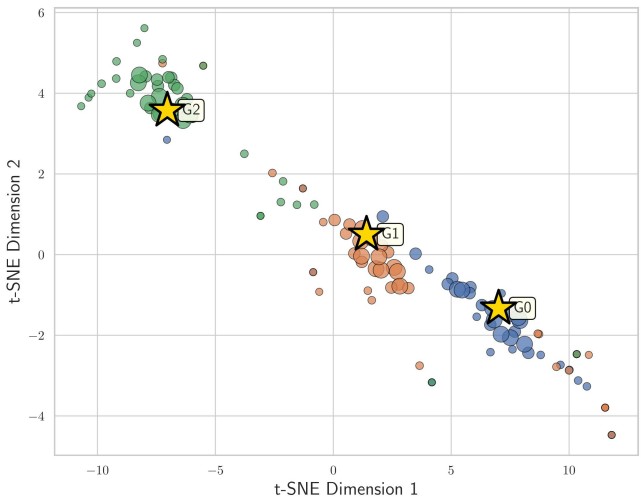

*Figure 5.* Low-dimensional projection of empirical moment vectors for graphs sampled from 3 graphons at node sizes $n \in \{5, 10, 100, 300\}$ (40 graphs per graphon per size). Spread shrinks and separation sharpens as $n$ grows.

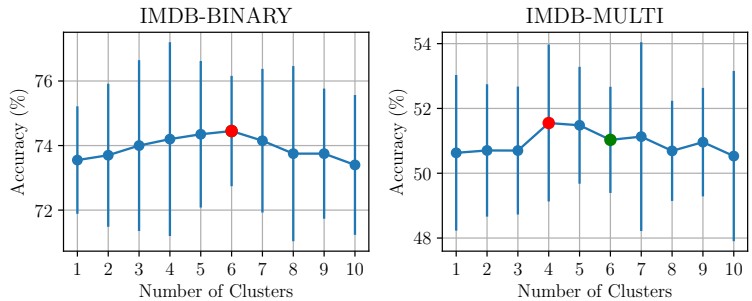

*Figure 6.* Effect of the number of clusters on GMAM performance.

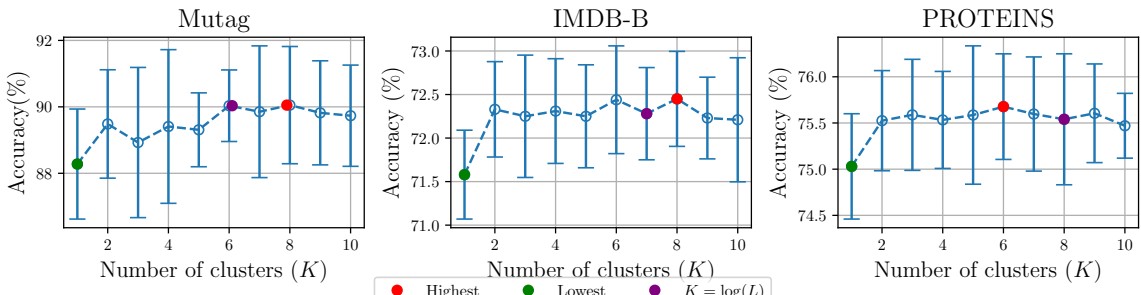

*Figure 7.* Effect of the number of clusters on MGCL performance.

TFR comparison across datasets

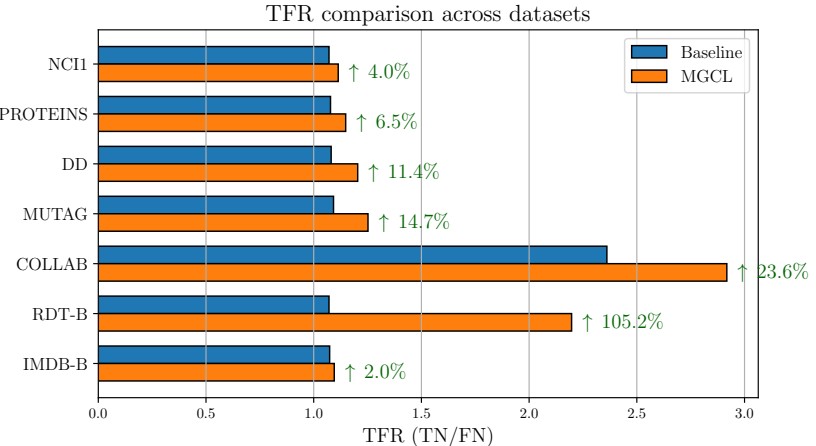

*Figure 8.* Effect of clustering on TFR across different datasets.

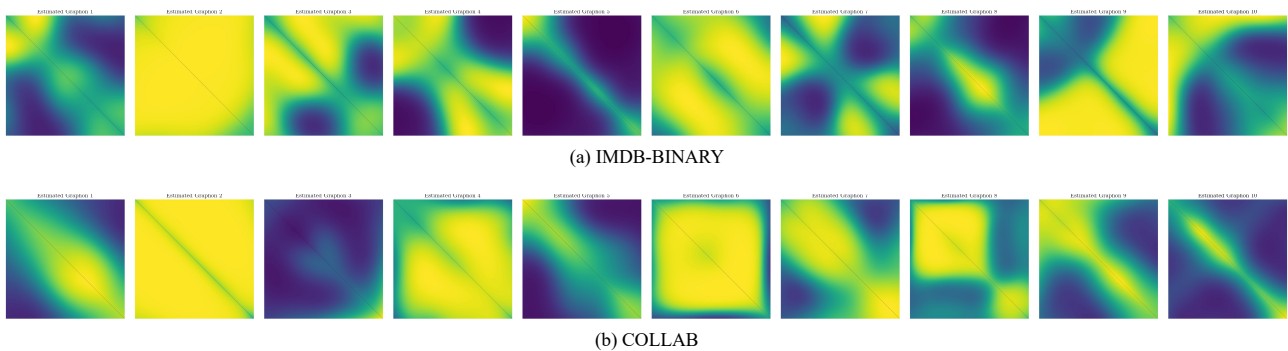

(a) IMDB-BINARY

(b) COLLAB

*Figure 9.* Cluster-specific estimated graphons in the COLLAB and IMDB-BINARY dataset, revealing diverse structures.

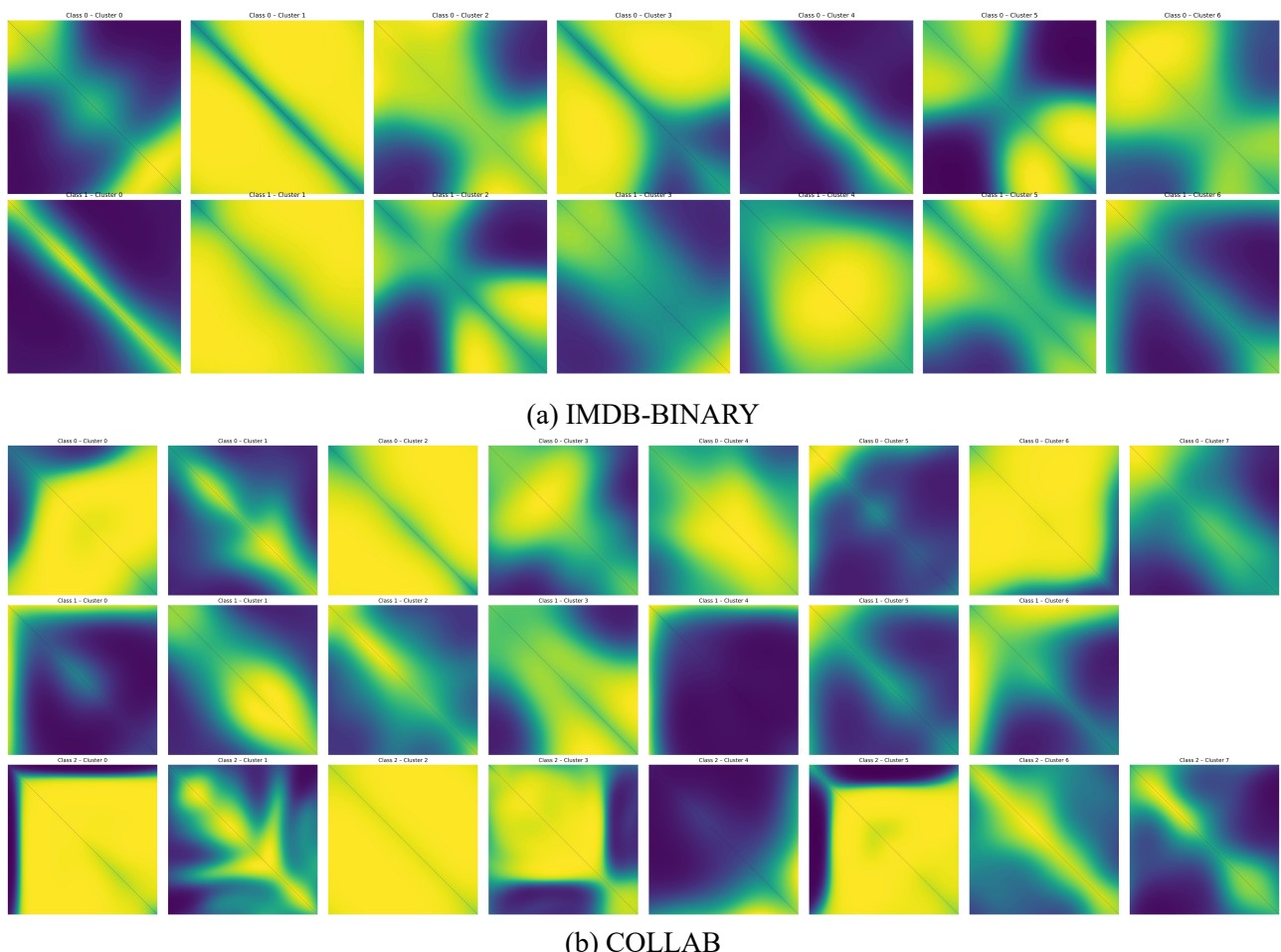

(a) IMDB-BINARY

(b) COLLAB

*Figure 10.* Cluster-specific estimated graphons in the COLLAB and IMDB-BINARY dataset within each class, revealing diverse structures.

