# OpenReview forum: "From Moments to Models: Graphon-Mixture Learning for Mixup and Contrastive Learning"
_ICML.cc/2026/Conference — ICML 2026 regular_

### Official Review · Reviewer_E8jJ · 2026-03-09

**Soundness:** 2
**Presentation:** 3
**Significance:** 2
**Originality:** 2
**Overall Recommendation:** 4
**Confidence:** 4

**Summary:**

The paper models graph datasets as mixtures of latent graphon generators. It first clusters graphs using motif-moment signatures, estimates mixture components, then uses that structure in two downstream methods: GMAM and MGCL. The paper includes theory, synthetic validation, and TU benchmark experiments.

**Compliance With Llm Reviewing Policy:**

Affirmed.

**Final Justification:**

After reading the rebuttal and the follow-up response. I still do not think the dense-graphon assumption is fully aligned with all of the benchmark settings, and I would like the final version to be very explicit that the theory currently covers a cleaner regime than some of the experiments. Also, the small/high-variance regime is still more clarified than fully stress-tested. So these concerns do not disappear. That said, I am now more persuaded that these are mainly scope limitations (but the author's latest response seems a little bit subjective and defensive) rather than direct contradictions of the method or the empirical results. Since the overall modeling story is coherent and the benchmark result seems strong, I am okay with supporting a borderline accept, while still noting the remaining caveats above.

**Key Questions For Authors:**

1. I think better to narrow the “first” claims with precise scope in related work and main claims.

2. Could you provide stronger stress tests for small/high-variance graph regimes where moment estimates are noisy?

3. Can you better isolate MGCL gain sources (negative-sampling rule vs augmentation rule)?

**Limitations:**

Partially yes. I think more explicit failure-mode discussion would help.

**Strengths And Weaknesses:**

**Strengths**
- I think the overall modeling story is coherent end-to-end.
- Theory and experiments are connected in a fairly meaningful way.
- Reported benchmark results are competitive and often strong.

**Weaknesses**
- I think some claim wording is a bit too strong (e.g., broad “first” style statements) relative to the evidence scope.
- The key assumption (moment vectors robustly reflect latent generators) can be fragile in smaller/noisier graphs; this is acknowledged, but not deeply stress-tested in main text.
- MGCL gains are not fully decomposed between better negative selection and graphon-aware augmentation, so mechanism clarity is still partial.

---

> ### Author Rebuttal · Authors · 2026-03-30
>
> We thank the reviewer for the constructive feedback. Below, we provide detailed responses to the raised concerns.
>
> **Response to W1 and Q1:**
> Our intended claim is not a broad “first” statement in graphon learning in general, but rather a more specific one: *to the best of our knowledge, this is the first work that explicitly estimates a *mixture of graphons* from a collection of observed graphs and then uses that mixture structure to improve *downstream graph learning tasks* (specifically, graph Mixup and graph contrastive learning).*
> Prior graphon-based works used in downstream learning, such as IGNR, SIGL, MomentNet, and G-Mixup, estimate or assume a *single* graphon for the observed data (or per class in supervised settings), rather than modeling the dataset as a mixture of multiple latent generative graphon components. Our contribution is precisely this additional mixture modeling step, and then leveraging it for model-aware augmentation and contrastive learning.
> We will revise this throughout the paper to narrow these claims accordingly, replacing broad “first” phrasing with more precise statements such as: *“to the best of our knowledge, this is the first framework for graphon-mixture estimation for downstream graph-level learning.”* We will also add explicit references to prior single-graphon methods (e.g., IGNR, SIGL, MomentNet, G-Mixup) in the related work and main claims to better clarify this distinction.
>
> **Response to W2 and Q2:**
> We thank the reviewer for raising this important question. Due to the character limit, we kindly refer the reviewer to the first part of our response to Reviewer **GDPf** (Response to W1 and Q3), where this concern is addressed.
>
> **Response to W3 and Q3:**
> This is a valid and fair point. We address this in two ways:
>
> *(1) Theoretical decomposition.*
> The lower bound in Theorem 3.2 consists of three terms:
> $\ln m_t + \mathrm{sim}(\mathbf{z}_t,\bar{\mathbf{z}}^{(\neg C_t)}) - \mathrm{sim}(\mathbf{z}_t,\tilde{\mathbf{z}}_t) \le \ell_t.$
> These terms separate the roles of graphon-informed augmentation(GIA) and cluster-aware loss(CAL). The alignment term $-\mathrm{sim}(\mathbf{z}_t,\tilde{\mathbf{z}}_t)$ depends on the positive pair and is affected by the augmentation strategy. With GIA, the augmented graph is generated by the same inferred graphon as the anchor, which yields a more semantically faithful positive sample than random perturbations. As discussed in Sec. 3.3 and supported by [1], this increases $\mathrm{sim}(\mathbf{z}_t,\tilde{\mathbf{z}}_t)$ and lowers the bound.
> CAL affects the remaining two terms. First, by restricting negatives to graphs from different clusters, reducing the $\ln m_t$ term compared to general $\ln N$. Second, excluding same-cluster graphs (which are likely false negatives) moves the negative centroid $\bar{\mathbf{z}}^{(\neg C_t)}$ farther from the anchor, reducing $\mathrm{sim}(\mathbf{z}_t,\bar{\mathbf{z}}^{(\neg C_t)})$.
>
> *(2) Empirical decomposition.*
> In addition to Figure 7, which isolates the false-negative reduction effect (independent of augmentation), we include an ablation here that separates the contributions of each component. Specifically, we consider the standard graph contrastive learning baseline GraphCL, and compare:
> 1. *GraphCL* = random augmentation + standard InfoNCE
> 2. *MGCL w/o CAL* = GraphCL + GIA
> 3. *MGCL* = GraphCL + GIA + CAL
>
> This isolates the contribution of GIA first, and then the additional contribution of CAL on top of it. On three datasets, the results show that GIA improves over GraphCL, and adding CAL on top of GIA yields a further consistent gain. We will include this ablation in the appendix to better isolate the mechanism behind MGCL’s improvement.
>
> | Method | PROT. | MUTAG | IMDB-B |
> |---|---:|---:|---:|
> | 1 | 74.39$\pm$.4 | 86.80$\pm$ 1.3 | 71.14$\pm$.4 |
> | 2 | 74.79$\pm$.6 | 88.29$\pm$ 1.7 | 71.38$\pm$.8 |
> | 3 | 75.54$\pm$.3 | 90.03$\pm$ 1.3 | 72.28$\pm$.5 |
>
> **Response to the limitation:** Thank you for this helpful suggestion. We will expand this section in the revision.
> In particular, we will add a clearer limitation discussion around the *small-graph regime*, where higher sampling error makes motif estimates less reliable and can reduce clustering/graphon-estimation quality. As discussed above, we will also support this with an additional experiment to more explicitly show this behavior.
> In addition, we will add a more explicit discussion of the *sparse and finite-graph setting*. Since graphons are most naturally suited to dense-graph asymptotics, we agree it is important to clarify this limitation in practice. We will discuss possible directions for improving robustness in such settings.
>
> We hope our clarifications and additional experiments adequately address the concerns raised and further strengthen the presentation and scope of the paper.
>
> [1] Ruiz et al. Graphon Neural Networks and the Transferability of Graph Neural Networks. NeurIPS, 2020.

---

> > ### Author Rebuttal · Reviewer_E8jJ · 2026-04-03
> >
> > Thank you for the response. I appreciate the narrower wording of the “first” claim and the additional MGCL component ablation; both are helpful. Still, my remaining concerns are only partially resolved. The discussion of the small/high-variance graph regime and the dense-graphon vs. sparse finite-graph mismatch is now clearer, but it is still mostly framed as clarification and planned revision rather than being fully addressed in the current paper. Since these points affect the scope and robustness of the proposed framework at a fairly core level, I do not think they are easy to settle within a short rebuttal. I therefore keep my original score unchanged.

---

> > > ### Author Response · Authors · 2026-04-04
> > >
> > > We thank the reviewer for the follow-up and for the time and effort spent engaging with our rebuttal. We are glad the 'first' claim clarification and MGCL ablation were appreciated and the discussion of the small/high-variance and dense-vs-sparse regimes is found clearer. That said, we respectfully disagree that these remaining concerns affect the *core* ideas of the work.
> > >
> > > - Small/high-variance graphs concern:
> > > The presence of small graphs does not affect the core methodology, either at the level of Theorem 3.1 or the pipeline design. The theorem already makes clear that smaller graphs have larger sampling variability, so this regime is naturally harder for **any method** and not a limitation specific to ours. Table 2 directly supports this: in the varying-size setting where small/high-variance graphs are present, our method still achieves substantially better clustering than all competing baselines. This is because our pipeline clusters based on motif structure and benefits from cluster-level aggregation rather than relying on individual graphs. The same holds on real-world datasets with small graphs (MUTAG, IMDB-B), where our method remains competitive or superior. We do not view the existence of small/high-variance graphs as a limitation of the core contribution, though we added the additional small-graph experiment as suggested by the reviewer to make this behavior explicit for the readers.
> > >
> > > - Dense-graphon vs. sparse finite-graph mismatch concern:
> > > We also do not view this as a limitation of the core methodology of the main novelty of the paper. Our framework is centered on motif-based identification of heterogeneous graph populations and leveraging that structure in downstream learning; this does not depend on the dense-graph asymptotic interpretation being exact for every finite sparse dataset. Moreover, graphons are widely used in practice as generative models for real-world graph data, including sparse settings, and a substantial body of prior work has applied graphon-based modeling and reported strong empirical performance [1–7]. Our own results on relatively sparse benchmarks are consistent with this.
> > >
> > > For these reasons, we respectfully disagree that these concerns affect the core scope or robustness of the proposed framework, or that they require a fundamental update for the work to be meaningfully built upon.
> > > Our work relying on graphons is not a limitation, as the positive empirical results indicate.
> > > Again, the asymptotic dense regime of graphons **does not invalidate their use to model sparse finite graphs**, and this has been shown in different fields in the literature across several application domains, like control and economics [1-3].
> > > Most importantly, we want to emphasize that our proposed modification is not a methodological change to the paper.
> > > We are not suggesting replacing graphons with another construction in the current paper, altering the graphon estimation procedure, or changing any component of the proposed pipeline, theory, or experiments.
> > > Rather, our proposed revision is simply to note this asymptotic behavior of graphons and to mention that the sparse nature of some real-world graphs can motivate the use of other random graph models that might be better suited for this data.
> > > This is a matter of framing and discussion, not a change to the fundamental methodology.
> > > Therefore, these concerns do not alter the core methodology, do not undermine the theory, and do not contradict the empirical evidence. They motivate a clearer framing of scope and assumptions, which we have already addressed through clarification and additional experiments during rebuttal, rather than changes to the contributions themselves.
> > >
> > > [1] Gao and Caines. Graphon Control of Large-Scale Networks of Linear Systems. IEEE Transactions on Automatic Control 2019.
> > >
> > > [2] Caines and Huang. Graphon Mean Field Games and Their Equations. SIAM Journal on Control and Optimization 2021.
> > >
> > > [3] Parise and Ozdaglar. Graphon Games: A Statistical Framework for Network Games and Interventions. Econometrica 2023.
> > >
> > > [4] Han et al. G-Mixup: Graph Data Augmentation for Graph Classification. ICML 2022.
> > >
> > > [5] Navarro and Segarra. GraphMAD: Graph Mixup for Data Augmentation Using Data-Driven Convex Clustering. ICASSP 2023.
> > >
> > > [6] Fabian et al. Learning Sparse Graphon Mean Field Games. AISTATS 2023.
> > >
> > > [7] Xia et al. Implicit Graphon Neural Representation. AISTATS 2023.
> > >
> > >
> > > **[Update] Response to the final justification:** We would like to sincerely thank the reviewer for taking the time to read our responses and for their continued engagement throughout the rebuttal process. We are glad our clarifications were helpful. We will gladly refer to your comments upon updating the manuscript to explicitly discuss the difference between the theoretical regime and some of the experimental settings, as well as the related scope limitations of our work.

---

### Official Review · Reviewer_NvDS · 2026-03-12

**Soundness:** 3
**Presentation:** 3
**Significance:** 2
**Originality:** 3
**Overall Recommendation:** 3
**Confidence:** 4

**Summary:**

This paper introduces a unified framework that models graph datasets as a mixture of multiple underlying probabilistic generative models (graphons). To estimate these mixture components, the authors propose representing graphs via their motif densities (graph moments) and clustering them using $k$-means. The theoretical core of the paper is Theorem 3.1, which provides a concentration bound demonstrating that graphs sampled from graphons with a small cut distance will exhibit similar motif densities. The extracted graphon mixture is then integrated into two downstream tasks: Graphon-Mixture-Aware Mixup (GMAM) for supervised data augmentation, and Model-aware Graph Contrastive Learning (MGCL) for unsupervised representation learning. The framework is evaluated on several TUDataset benchmarks.

**Compliance With Llm Reviewing Policy:**

Affirmed.

**Final Justification:**

Thank you to the authors for providing a detailed rebuttal and for the clarifications offered in your response. I have carefully reviewed your explanations.

However, my primary reservations regarding the core methodology remain unresolved. Specifically, I am still unconvinced of the practical significance of the mixed data obtained through the Graphon-based Mixup method. It remains highly ambiguous whether interpolating between latent graphons produces semantically meaningful structural information for real-world graph topologies, rather than simply generating arbitrary structural noise.

Additionally, the evidence provided does not sufficiently prove the true empirical effectiveness of this approach in practical scenarios. Because these fundamental concerns about the method's practical utility and effectiveness persist, I am maintaining my negative rating.

**Key Questions For Authors:**

1. Could you provide a deeper theoretical motivation for why truncated motif densities can be safely used as an *equivalent* for separating graphon-based mixtures, and under what structural conditions this equivalence might fail?
2. Are graphon-based mixtures universally applicable to all graph data, or is their theoretical and empirical validity restricted only to graph data with specific distributions or specific evolutionary scenarios (e.g., dense graphs)?
3. What exact problem can the proposed method solve that simpler latent space clustering cannot? Furthermore, are there similar problems in real-world scenarios that explicitly necessitate modeling data as a mixture of continuous graphons?

**Limitations:**

Yes

**Strengths And Weaknesses:**

# Strengths

* Moving away from the assumption that an entire dataset or class is generated by a single monolithic graphon is a conceptually sound direction. Explicitly modeling structural heterogeneity as a mixture of generative processes is an intuitive approach to mitigating noisy augmentations.
* The derivation of a tighter sampling error bound for empirical motif densities in Theorem 3.1 is mathematically rigorous and offers a meaningful improvement over classical single-stage bounds.

# Weaknesses

* The algorithm's foundational motivation is not clearly explained. While the paper bounds the difference in empirical densities via cut distance, it fails to rigorously justify why truncated motif densities (specifically up to 4 nodes) can be used as a fully *equivalent* and sufficient representation for disentangling complex, infinite-dimensional graphon-based mixtures.
* The paper does not adequately define the theoretical boundaries of its framework. Graphons are classically defined as the limit objects of dense graph sequences. It remains highly ambiguous whether the proposed graphon-based mixtures are universally applicable to *all* graph data (e.g., sparse networks, heterophilic graphs, or scale-free networks), or if their validity is strictly confined to graph data with specific dense distributions or scenarios.
* The methodological complexity of estimating graphon mixtures is presented without a compelling, concrete use-case. What specific problem can the proposed method solve that standard latent representation clustering cannot? The manuscript lacks a discussion on whether there are actual, similar problems in real-world scenarios where datasets are explicitly generated from a discrete mixture of distinct graphons, making the practical utility of the framework questionable.
* The empirical evaluation relies entirely on small-scale, saturated datasets (e.g., MUTAG, PROTEINS, DD). Evaluating representation learning and data augmentation strategies exclusively on these toy-scale datasets is insufficient to prove that the complex mixture modeling provides meaningful regularization for deep architectures in modern scenarios.

---

> ### Author Rebuttal · Authors · 2026-03-30
>
> We thank the reviewer for their thorough review. Below, we provide detailed responses to the raised concerns.
>
> **W1,Q1:** We agree that truncated motif densities should *not* be interpreted as a fully equivalent finite-dimensional representation of an arbitrary infinite-dimensional graphon. Our claim is more modest. They serve as a *practical, theoretically motivated proxy* for separating graphon-based mixtures under finite-sample constraints.
> Theorem 3.1 provides the key motivation that for any fixed motif family, closeness in cut distance implies closeness of empirical motif-density vectors with high probability. This justifies motif-based clustering as a stable surrogate for separating latent components, but also reveals a clear bias-variance trade-off. As the motif size $k$ increases, the feature space becomes richer, but the sampling error also increases in finite graphs. In our bound, the effective number of independent motif placements decreases with $k$, so larger motifs yield noisier estimates. Thus, higher-order motifs are not automatically better in finite-sample settings.
> This is why we use a truncated set of connected motifs up to 4 nodes in the main experiments as a bias-variance trade-off, not a claim of exact sufficiency. This is also supported by the ablation in App. F.1 (Tab. 7), where we extend the feature set to motifs up to 5 nodes. Clustering accuracy improves quickly at first, then largely plateaus; after the first 9 motifs (all up to 4 nodes), additional motifs provide only marginal gains. This suggests that the truncated motif set captures most of the useful discriminative signal in our setting while avoiding unnecessary variance and complexity.
> Broadly, this approximation can fail when distinct graphons are indistinguishable under the chosen truncated motif family (agreeing on low-order motifs but differing in higher-order structure), or in the small-graph regime where sampling noise obscures separation. We will make these failure modes more explicit in the revision.
> Finally, our empirical results support that the chosen truncation is sufficient for the target task, rather than universally sufficient in theory. In Tab.2, moment-based clustering nearly matches the theory-based upper bound and substantially outperforms alternative embedding-based baselines on the synthetic graphon-mixture setting.
>
> **W2,Q2:** We thank the reviewer for raising this important point. Due to the character limit, we kindly refer the reviewer to our response to Reviewer **mh2B** (W1,Q), where we address this closely related concern.
>
> **W3,Q3:** We would like to clarify that we do *not* claim real-world graph datasets are generated by an exact discrete mixture of graphons. Rather, we view a graphon mixture as a *useful nonparametric generative approximation* for heterogeneous graph collections, analogous to how mixture models (e.g., Gaussian mixtures) are used in practice as flexible approximations rather than exact data-generating models. Our point is that heterogeneous graph datasets can often be partitioned into groups, each with a latent generator producing graphs with similar structural footprints, as reflected in their moments. This distinguishes our method from standard latent-space clustering, which may group graphs, but it typically does *not* provide an explicit graph generator for each cluster. In contrast, our framework jointly provides: (i)clustering into latent components, and (ii)an estimated graphon for each component, which can be directly used for *generation/resampling*. This distinction is reflected in Tab.2, where alternative clustering/embedding methods are less effective at recovering the latent generative structure. This is exactly the capability needed in GMAM and MGCL, where the goal is not only to group similar graphs but also to generate semantically faithful augmentations.
> Finally, our method is also theoretically grounded (Theorem 3.1), rather than being only a heuristic clustering pipeline. This gives a principled reason to prefer motif-based graphon-mixture estimation over arbitrary latent embeddings when the goal is to recover *generative components*.
>
> **W4:** We follow the standard protocol in the mixup and contrastive learning literature, using established TUD benchmarks to enable fair and direct comparison with prior methods in the same setting. This is important here, since we aim to evaluate whether graphon-mixture modeling improves existing pipelines such as GMixup and GraphCL, and to compare against recent SOTA methods (App. E.4).
> Also, our evaluation is not limited to only the smallest datasets. Beyond MUTAG, PROT, and DD, we also include larger benchmarks such as RDT-B,M5,M12 and COLLAB. For graph-level tasks, dataset scale is also determined by the number of graphs, and these benchmarks include large graph collections.
>
> We hope our responses help address the concerns raised and better clarify the motivation, scope, and practical relevance of our framework.

---

> > ### Author Rebuttal · Reviewer_NvDS · 2026-04-03
> >
> > Thank you to the authors for their efforts in preparing the rebuttal and clarifying aspects of the manuscript. However, after carefully reading the response, my core concerns regarding the practical utility and empirical validation of the proposed method remain unresolved.
> >
> > Specifically, I still have significant reservations about the Graphon-Mixture-Aware Mixup (GMAM) method. What is the actual practical significance of the mixed graph data generated through this graphon-based interpolation? In real-world domains—such as molecular chemistry or complex social networks—graph topologies are highly discrete and governed by strict, domain-specific rules. While interpolating between latent graphons might produce mathematically valid intermediate probability matrices, it is highly questionable whether the graphs sampled from these "mixed" graphons carry any semantically meaningful structural information. There is a strong likelihood that this approach merely generates arbitrary structural noise rather than useful, representative augmented data.
> >
> > Furthermore, I remain unconvinced of its true empirical effectiveness. Demonstrating marginal performance gains primarily on toy-scale, saturated benchmarks (such as the TUDatasets) does not sufficiently prove that graphon-based mixup provides meaningful regularization or generalization benefits for complex, large-scale graph machine learning tasks where data augmentation is actually critical.
> >
> > Because the practical significance of the generated mixed data remains fundamentally ambiguous, and the empirical validation is not robust enough to dispel these reservations, I do not believe the method is ready for deployment in real-world scenarios. Therefore, I am maintaining my original score based on the current version of the manuscript.

---

> > > ### Author Response · Authors · 2026-04-05
> > >
> > > While we appreciate the concern and agree that, in real-world graph domains with discrete and structured topologies, unconstrained interpolation could in principle produce weak/implausible augmentations, we believe this does not reflect a limitation of our work, but rather raises a broader question about the validity of graphon-based augmentation itself.
> > >
> > > Our contribution is not to newly justify graphon-based augmentation in full generality, but to address an important practical weakness of prior graphon-based methods such as[1,2], namely the assumption that each class can be represented by a single graphon.Prior work, notably G-mixup[1], has shown that graphon-based augmentation can improve *generalization, training stability, and robustness* of GNNs, without requiring that each individual augmented graph be interpreted as a strictly valid domain-specific sample in isolation. This is standard in mixup-style augmentation, where interpolated samples primarily serve as regularizers between observed structures rather than as fully meaningful objects on their own. The situation is similar to image mixup, where the mixed image is not necessarily semantically meaningful as a standalone sample[3,4]. Moreover, the theoretical analysis in [1] shows that interpolation between graphons can preserve discriminative motifs from the original graphons in the mixed graphon, supporting that graphons can serve as a meaningful latent model for interpolation and augmentation, and the generated graphs are not just arbitrary structural noise.
> > > Thus, whether graphon-based interpolation can produce useful augmentations *is a premise established in prior work,* and not introduced by our paper.  We are motivated by a practical limitation of this premise in real-world settings: modeling each class with a single graphon can be too coarse when the class contains multiple latent structural modes. This aligns with the reviewer’s concern that if the underlying generator is overly simplistic, interpolation may become less semantically meaningful. Our contribution is to address this by replacing the single graphon with a mixture of latent generators, thereby addressing a weakness of graphon-based augmentation in real data.
> > >
> > > This is also supported empirically.If generated graphs were primarily arbitrary structural noise, one would expect degraded downstream performance, especially relative to the vanilla baseline (no augmentation). Table 1 shows that this is not the case, and highlights the limitation of the single-graphon assumption. Compared to the vanilla baseline, G-Mixup improves performance on the first four datasets but does not outperform vanilla on the last three. In contrast, GMAM improves over G-Mixup on all datasets, and importantly, also outperforms vanilla on the last three datasets.
> > > This suggests that the issue is not graphon-based augmentation itself, but the use of an overly coarse single-graphon model. By replacing it with a mixture of latent generators,GMAM produces more faithful augmentations, especially where single-graphon augmentation appears to break down.
> > > To further test the molecular chemistry setting, we added a simple experiment on the AIDS dataset by visualizing real and GMAM-augmented graphs in a lower-dimensional projection of the motif-vector space in this **[plot](https://imgur.com/a/7gN3OWZ)**.The augmented samples remain close to the real data distribution, providing additional evidence that GMAM is not generating arbitrary structural noise even in this molecular setting. Combined with Table 1, this further supports that graphon-mixture augmentation produces useful and more faithful synthetic data.
> > > Regarding the empirical effectiveness, our goal is not to claim that GMAM is a universal augmentation method for all graph domains, but rather to show that, under the accepted evaluation setting in this literature, *mixture-aware graphon* modeling can improve augmentation and downstream performance. Accordingly, using TUDatasets follows the accepted benchmark protocol in graph mixup, including prior graphon-based methods.
> > >
> > > Therefore,we respectfully believe the relevant question for GMAM is not whether graphon-based interpolation is meaningful in general-that has been supported by prior work-but rather whether graphon mixture estimation *improves* upon single-graphon augmentation.
> > > Whether generative models beyond graphons (diffusion models respecting the discrete boundaries of graph data) could improve augmentation is an interesting direction, but it is a broader research question beyond the scope and motivation of our work, and would require its own theoretical grounding.
> > >
> > > [1] Han et al. G-Mixup ICML 2022
> > >
> > > [2] Navarro Segarra GraphMAD ICASSP 2023
> > >
> > > [3] Tokozume et al. Between-Class Learning for Image Classification CVPR 2018
> > >
> > > [4] Cao et al. A Survey of Mix-Based Data Augmentation ACM 2024
> > >
> > > **[Update]** We sincerely appreciate the reviewer’s time, follow-up, and continued engagement throughout the rebuttal process.

---

### Official Review · Reviewer_mh2B · 2026-03-12

**Soundness:** 3
**Presentation:** 3
**Significance:** 3
**Originality:** 3
**Overall Recommendation:** 5
**Confidence:** 4

**Summary:**

This paper proposes a framework for modeling dense graph datasets as samples from a mixture of graphon-based generative models. The core motivation is that real-world graph data exhibits population variability better captured by a combination of distributions than by a single graphon. The authors develop an estimation method based on homomorphism densities, combining clustering with an implicit neural representation, and derive theoretical guarantees relating graphon structural similarity to motif density similarity. The framework is instantiated in two downstream settings: graph data augmentation (GMAM) and contrastive learning (MGCL).

**Compliance With Llm Reviewing Policy:**

Affirmed.

**Final Justification:**

I retain my original score (accept).

**Key Questions For Authors:**

- Have the authors considered discussing how the dense graph assumption affects results on sparse benchmarks, even informally?

**Limitations:**

No. See weaknesses above.

**Strengths And Weaknesses:**

Strengths

- The paper is well-motivated and theoretically grounded. The hypothesis that real-world graph datasets arise from mixtures of generative models is well-justified, and the novel guarantees connecting graphon structure to motif densities are a meaningful contribution.
- The empirical validation is thorough. Ablation studies cover robustness to the number of mixture components, sufficiency of small motifs (up to 4 nodes), and the advantage of mixture modeling over single-graphon baselines for clustering quality.
- The framework integrates cleanly into established pipelines, demonstrating practical value through two concrete applications.

Weaknesses

- The paper does not adequately discuss the limitations of graphon-based modeling. The dense graph assumption is never made explicit, yet several benchmarks (e.g., NCI1, MUTAG) contain sparse graphs. The authors should address this gap and discuss its implications for empirical performance.
- The discussion of Theorem 3.1 (lines 193–210) contains an imprecision. Lines 196–202 suggest a bidirectional relationship between cut distance and motif density similarity, but this does not hold in general. Graphs with dissimilar motif densities can still arise from graphons that are close in cut distance---small graphs sampled from the same graphon are a straightforward counterexample, and the synthetic experiments (graphons 2 and 3) already hint at this.

---

> ### Author Rebuttal · Authors · 2026-03-30
>
> We appreciate the reviewer’s feedback and the opportunity to clarify these points. Below, we provide detailed responses to the raised concerns.
>
> **Response to W1 and Q:**
> We thank the reviewer for raising this important point. We agree that the current paper is developed in the classical dense-graphon setting, and we will make this scope more explicit in the revision. Accordingly, the present theory is not intended to fully cover highly sparse large-scale networks, and graphon-based mixtures should not be interpreted as universally valid for all graph data without qualification.
>
> That said, we view this as a limitation of the particular latent graph model adopted in the current paper, rather than of the overall framework. In our approach, the graphon primarily serves as a cluster-level latent generator, while the key contribution is the motif-based discovery of heterogeneous graph populations. When the data are sparse, this latent model can in principle be replaced by a sparse-network analogue. At the same time, even when the true data are sparse, a learned dense graphon can still provide a useful cluster-level approximation by matching the relevant motif (or moment) structure and generating graphs with similar aggregate characteristics.
>
> This interpretation is consistent with our empirical findings. We agree that the dense-graph assumption should have been stated more explicitly, especially since several benchmark datasets are sparse (e.g., NCI1 and MUTAG). Informally, we believe this mainly affects the modeling interpretation of the latent generator, rather than the usefulness of the overall pipeline. In particular, the main practical benefit of our method comes from motif-based identification of latent subpopulations, and this mechanism can remain effective even when the dense graphon is used as an approximation. Consistent with this, the method still performs well on sparse benchmarks, suggesting that the estimated dense graphons still capture useful cluster-level structure in these settings.
> From a theoretical perspective, this can also be relaxed. For sparse random graph models of the form $\rho_n W$, one can work with suitably normalized motif densities, and the same decomposition underlying our theorem continues to hold with sparsity-dependent concentration terms. This aligns with the standard $L^p$ sparse graphon literature [1,2], while $L^p$ graphons and graphex-type models provide natural extensions beyond the dense setting [1,2,3]. We will add this discussion and the corresponding references in the revision.
>
> We therefore view the current dense-graph formulation as a clean and theoretically tractable first setting, rather than the only regime where the framework applies. At the same time, we agree that a sparse latent model and corresponding sparse graphon estimation procedure would provide a better theoretical and modeling match for sparse datasets, and could potentially improve performance further. We will make this limitation and future direction more explicit in the revision.
>
> **Response to W2:** This is a very nice observation, and we thank the reviewer for pointing it out.
> Theorem 3.1 only guarantees that closeness in cut distance implies closeness of empirical moment vectors up to a sampling error term; the converse does not hold unconditionally in finite samples, since small graphs sampled from the same (or very similar) graphon can still have noticeably different empirical motif densities due to sampling variability.
> We will revise the text accordingly. In particular, we will replace the sentence with a more precise statement such as: *"Alternatively, when the sampling error is sufficiently small (e.g., for sufficiently large graphs), a large observed discrepancy between the empirical moment vectors of two graphs provides evidence that their underlying graphons are not close in cut distance."*
> This better reflects the scope of Theorem 3.1 and avoids overstating the converse in the finite-sample regime.
>
>
> We thank the reviewer again for their constructive feedback and positive assessment. We hope our clarifications adequately address the concerns.
>
> [1] Borgs et al. *An $L^p$ Theory of Sparse Graph Convergence I: Limits, Sparse Random Graph Models, and Power Law Distributions*. *Trans. Amer. Math. Soc.*, 2019.
>
> [2] Borgs et al. *An $L^p$ Theory of Sparse Graph Convergence II: LD Convergence, Quotients and Right Convergence*. *Ann. Probab.*, 2018.
>
> [3] Borgs et al. *Sparse Exchangeable Graphs and Their Limits via Graphon Processes*. *JMLR*, 2018.

---

> > ### Author Rebuttal · Reviewer_mh2B · 2026-04-02
> >
> > I retain my original score (accept).

---

> > > ### Author Response · Authors · 2026-04-03
> > >
> > > We thank the reviewer for the encouraging assessment of our work and for the helpful suggestions to further strengthen the manuscript.

---

### Official Review · Reviewer_GDPf · 2026-03-13

**Soundness:** 3
**Presentation:** 4
**Significance:** 3
**Originality:** 4
**Overall Recommendation:** 5
**Confidence:** 4

**Summary:**

This work proposes a unified framework to model graph data as a mixture of underlying graphons. The authors present a principled and mathematically grounded framework to handle heterogeneity in graph data.

**Compliance With Llm Reviewing Policy:**

Affirmed.

**Final Justification:**

The authors have resolved my concerns. I'd like to keep my positive rating.

**Key Questions For Authors:**

1. In practice, is there a principled way to determine the number of underlying generative models? i.e. K?
2. As graphons are limit models for dense graphs, is there a way to relax this and include sparse or relatively dense graphs?
3. In theorem 3.1, the bound decrease with graph size n, will the sampling error be too large for small graphs and could you comment on this case?

**Limitations:**

Yes

**Strengths And Weaknesses:**

Strengths:
1. The graphon-mixture model is mathematically robust to bridge the gap between classical graph limit and the practical heterogeneous graph datasets.
2. The authors propose a novel two-stage analysis of sampling error and achieve tighter concentration bounds.
3. The computation is efficient with ORCA algorithm according to the runtime analysis.

Weakness:
1. As the theory shows the concentration of motif densities, small graphs will have larger sampling noise, which may lead to larger errors.
2. Graphon has the limitation to model dense graphs and fail to capture large scale real world sparse graphs.
3. The success of the downstream tasks depend heavily on the quality of the graphon estimators.

---

> ### Author Rebuttal · Authors · 2026-03-30
>
> We thank the reviewer for their thoughtful evaluation of our work. Below, we provide detailed responses to the raised concerns.
>
> **W1,Q3:**
> It is true that Theorem 3.1 explicitly indicates that the finite-sample error is larger for smaller graphs. In particular, in Eq.(6), the second term corresponds to the sampling error, and this term decreases with the graph size $n$. Therefore, for very small graphs, the empirical motif densities are naturally noisier and may deviate more from their graphon-level expectations. This is the regime where one should expect weaker concentration and, consequently, potentially larger clustering or estimation errors. That said, this behavior is discussed and illustrated in the paper. In Fig.2(a), where graphs have varying sizes, the clusters are visibly more diffuse, which is consistent with the larger sampling variability expected for smaller graphs. In contrast, in Fig.2(b), when the graph size is fixed at $n=200$, the empirical moment vectors concentrate much more tightly around their corresponding graphon means, producing cleaner separation. Hence, both the theorem and the synthetic experiment support the conclusion that smaller graphs are noisier, while larger graphs benefit from stronger concentration. Although we agree that the sizes of individual graphs are not explicitly shown for each datapoint in Fig.2. To examine this point directly and make it clear, we additionally ran a similar experiment on 3 graphons using 40 sampled graphs from each graphon at node sizes $\\{5,10,100,300\\}$. Plot in this anonymous **[link](https://imgur.com/a/tMp4zz4)**. For very small graphs, the moment vectors exhibit noticeably larger spread, whereas for larger graphs they concentrate more tightly around the graphon-specific centers, leading to clearer separation. We will add a subsection in the appendix to add this clarifying experiment in the revision.
> There is also a complementary message from the graphon estimation side. In MomentNet [1], the empirical moment vector is formed by averaging motif-density vectors across multiple graphs, and both their theory and experiments show that estimation improves as either the graph size or the number of graphs increases. Even in relatively small or high-variance settings, this averaging reduces variance and enables accurate recovery of the underlying graphon family. This is closely aligned with our setting. Our framework does not rely on a single graph in isolation, but instead clusters graphs in moment space and estimates a representative latent generator from groups of similar graphs. As a result, it benefits from a cluster-level averaging effect, which helps mitigate noise in individual small graphs and helps explain the positive empirical performance even when some benchmark graphs are relatively small. We agree that this regime deserves more discussion, and we will clarify that robustness depends more strongly on the number of available graphs and the stability of the estimated motif statistics.
>
> **W2,Q2:**
> We thank the reviewer for raising this important point. Due to the character limit, we kindly refer the reviewer to our response to Reviewer **mh2B** (W1,Q), where we address this closely related concern in detail, including both the theoretical scope of the current dense-graphon setting and natural extensions to sparse graphon-type models.
>
> **W3:**
> This is a valid point, and it is why we use SIGL in our experiments: it is a strong graphon estimator and provides access to node-level latent variables. At the same time, our framework is modular and not tied to a single estimator. Alternatives such as MomentNet or IGNR can be used in other regimes where such latents are not needed; in particular, MomentNet is especially appealing for small graphs due to its scalability and runtime. More generally, improved or better-adapted estimators (e.g., for sparse regimes) can be directly integrated into our pipeline and may further improve downstream performance. We will clarify this in the revision.
>
> **Q1:**
> In the current paper, we do not propose a automatic procedure for selecting $K$, and we cluster the motif-density embeddings using $k$-means and set the number of clusters via the heuristic $K=\log T$, where $T$ is the number of graphs. The appendix shows that the method is reasonably stable over a meaningful range of $K$, although $K$ remains a tunable hyperparameter depending on dataset heterogeneity. More broadly, determining $K$ boils down to the same model-selection problem that appears in standard clustering methods, and one could incorporate standard cluster-number selection strategies in the embedding space as a natural extension, for example, the elbow method. We will clarify this point in the revision.
>
> We thank the reviewer again for their constructive feedback. We hope our clarifications adequately address the raised concerns.
>
> [1] Ramezanpour et al. A few moments please: Scalable graphon learning via moment matching. NeurIPS 2025

---

> > ### Author Rebuttal · Reviewer_GDPf · 2026-04-03
> >
> > The authors have solved my questions and I would like to keep my positive score.

---

> > > ### Author Response · Authors · 2026-04-03
> > >
> > > We thank the reviewer for the positive assessment of our work and for the valuable feedback and suggestions for improving the manuscript. We are glad that the rebuttal was helpful.

---

### Decision · Program_Chairs · 2026-04-30

**Decision:**

Accept (regular)

**Comment:**

The submission received overall positive reviews, with reviewers generally recognizing the problem and the modeling direction as
as interesting. Reviewers appreciated the principled attempt to bridge graphon-based modeling with practical graph representation learning, as well as the combination of theoretical motivation and competitive experiments.

The rebuttal discussion improved clarity about the qualities of the work. It clarified several important points about the intended scope of the contribution, the role of the graphon-mixture interpretation, the use of moments for clustering, and the empirical motivation for the proposed augmentation and contrastive components. This led some reviewers to raise their positive assessment, and even where opinions remained somewhat split, the overall direction of the discussion was that the paper became clearer and better justified after rebuttal.

Some reservations remained at the end of the discussion. The main one concerned the dense-graphon interpretation and, more broadly, the exact breadth of the theoretical claims relative to sparse graph settings and practical graph regimes. However, this issue was not developed into a decisive objection in the internal discussion, nor was it substantiated to the point of undermining the main empirical and methodological contribution. I therefore support the overall positive assessment of the submission.